# Therapeutic targeting of measles virus polymerase with ERDRP-0519 suppresses all RNA synthesis activity

**Robert M. Cox**[1☯]**, Julien Sourimant**[1☯]**, Mugunthan Govindarajan**[2]**, Michael G. Natchus**[2]**, Richard K. Plemper**[1]*

**1** Institute for Biomedical Sciences, Georgia State University, Atlanta, Georgia, United States of America,
**2** Emory Institute for Drug Development, Emory University, Atlanta, Georgia, United States of America

☯ These authors contributed equally to this work.
* rplemper@gsu.edu

**Data Availability Statement:** All relevant data are within the manuscript and its Supporting Information files.

## Abstract

Morbilliviruses, such as measles virus (MeV) and canine distemper virus (CDV), are highly infectious members of the paramyxovirus family. MeV is responsible for major morbidity and mortality in non-vaccinated populations. ERDRP-0519, a pan-morbillivirus small molecule inhibitor for the treatment of measles, targets the morbillivirus RNA-dependent RNA-polymerase (RdRP) complex and displayed unparalleled oral efficacy against lethal infection of ferrets with CDV, an established surrogate model for human measles. Resistance profiling identified the L subunit of the RdRP, which harbors all enzymatic activity of the polymerase complex, as the molecular target of inhibition. Here, we examined binding characteristics, physical docking site, and the molecular mechanism of action of ERDRP-0519 through label-free biolayer interferometry, photoaffinity cross-linking, and *in vitro* RdRP assays using purified MeV RdRP complexes and synthetic templates. Results demonstrate that unlike all other mononegavirus small molecule inhibitors identified to date, ERDRP-0519 inhibits all phosphodiester bond formation in both *de novo* initiation of RNA synthesis at the promoter and RNA elongation by a committed polymerase complex. Photocrosslinking and resistance profiling-informed ligand docking revealed that this unprecedented mechanism of action of ERDRP-0519 is due to simultaneous engagement of the L protein polyribonucleotidyl transferase (PRNTase)-like domain and the flexible intrusion loop by the compound, pharmacologically locking the polymerase in pre-initiation conformation. This study informs selection of ERDRP-0519 as clinical candidate for measles therapy and identifies a previously unrecognized druggable site in mononegavirus L polymerase proteins that can silence all synthesis of viral RNA.

## Author summary

The mononegavirus order contains major established and recently emerged human pathogens. Despite the threat to human health, antiviral therapeutics directed against this order remain understudied. The mononegavirus polymerase complex represents a

**Funding:** This work was supported, in part, by Public Health Service grants AI071002 (to R.K.P.) and AI153400 (to R.K.P.), from the NIH/NIAID. The funders had no role in study design, data collection and analysis, decision to publish, or preparation of the manuscript.

**Competing interests:** I have read the journal's policy and the authors of this manuscript have the following competing interests: R.K.P. is an inventor on patent application PCT/US2012/030866, which includes the structure and method of use of ERDRP-0519. This study could affect his personal financial status. All other authors declare no competing interests.

promising drug target due to its central importance for both virus replication and viral mitigation of the innate host antiviral response. In this study, we have mechanistically characterized a clinical candidate small-molecule MeV polymerase inhibitor. The compound blocked all phosphodiester bond formation activity, a unique mechanism of action unlike all other known mononegavirus polymerase inhibitors. Photocrosslinking-based target site mapping demonstrated that this class-defining prototype inhibitor stabilizes a pre-initiation conformation of the viral polymerase complex that sterically cannot accommodate template RNA. Function-equivalent druggable sites exist in all mononegavirus polymerases. In addition to its direct anti-MeV impact, the insight gained in this study can therefore serve as a blueprint for indication spectrum expansion through structure-informed scaffold engineering or targeted drug discovery.

## Introduction

Morbilliviruses belong to the *Paramyxoviridae* family of highly contagious respiratory RNA viruses with negative polarity genomes. The archetype of the morbillivirus genus, MeV, is the most infectious pathogen identified to date with primary reproduction rates of 12–18 [1,2]. Although measles is a vaccine-preventable disease, MeV remains responsible for approximately 100,000 deaths annually worldwide and endemic transmission persists in large geographical regions. Due to its exceptional contagiousness, MeV is typically the first pathogen to reappear when vaccination coverage drops in an area [3,4]. In the aftermath of parental concerns about vaccination safety, four European countries, Albania, Czechia, Greece and the United Kingdom, have regressed to pre-measles eradication status [5]. A massive surge in global measles cases is feared as a result of the SARS-CoV-2 pandemic, since immunization campaigns have been suspended as part of the COVID-19 response [6–8]. Intensifying preparedness to mitigate the mounting problem is imperative. Effective anti-MeV therapeutics may aid by providing avenues to improved disease management and rapid outbreak control [9].

Towards identifying applicable measles therapeutics, we have developed a MeV inhibitor from high-throughput screening hit to orally bioavailable clinical candidate [10–13]. The optimized lead compound, ERDRP-0519, showed pan-morbillivirus inhibitory activity [10], which opened an opportunity for informative efficacy testing in ferrets infected with canine distemper virus (CDV) as a natural animal host surrogate assay for human morbillivirus disease [14]. CDV infection of ferrets recapitulates the hallmarks of human measles, but is invariably lethal within 10 to 14 days of infection, providing a definitive efficacy endpoint [14]. Initiating oral treatment of ferrets with ERDRP-0519 at the first day of viremia reversed infection outcome, however: all treated animals survived, clinical signs were alleviated, and recoverees mounted a robust immune response protecting against re-infection [10].

Mechanistically, ERDRP-0519 blocks morbillivirus polymerase activity [10,11]. Conserved among all mononegaviruses, paramyxoviruses encode RdRP complexes that consist of a large (L) polymerase protein harboring all enzymatic activities and a mandatory chaperone, the viral phosphoprotein (P) [15–19]. In addition to phosphodiester bond formation, L is considered to mediate capping of viral mRNAs through polyribonucleotidyltransferase (PRNTase) and methyltransferase (MTase) activities [16,20]. Although P does not contain enzymatic activity of its own, it is an essential cofactor required for proper L folding and establishing physical contact between the P-L polymerases and the viral genome, which consists of a ribonucleoprotein (RNP) complex of genomic RNA encapsidated by the viral nucleocapsid (N)

protein [21,22]. Several structural reconstructions of mononegavirus polymerase have been reported recently that provided essential insight into the organization of rhabdovirus (vesicular stomatitis virus and rabies virus [23,24]), pneumovirus (RSV [25]), and paramyxovirus (parainfluenzavirus 5 (PIV-5) [26]) polymerase complexes.

All reconstructions show an N-terminal RdRP domain (Fig 1A and 1B) composed of finger, palm, and thumb subdomains as present in other viral polymerases that contains six (A-F) structural motifs (Fig 1C) [27]. Of these, motif C harbors broadly conserved GDNQ residues that are considered to form the core catalytic site for phosphodiester bond formation [27,28]. Both the RdRP and adjacent PRNTase domains assume a ring-like structure, in which the GDNQ catalytic loop is oriented towards a large central cavity (Fig 1B and 1C). The predicted PRNTase domain contains an additional five sequence elements (PRNTase motifs A-E) that are highly conserved [27]. The histidine moiety of histidine-arginine (HR) duplet in PRNTase motif D specifically forms a transient covalent bond with the first transcribed nucleotide of a nascent RNA strand [27,29,30], which is subsequently resolved through a nucleophilic attack by GDP, initiating RNA cap formation [31]. In the PIV-5 L structure, motif D is located immediately adjacent to a flexible intrusion loop, which occupies the active site of the polymerase (Fig 1C). Based on comparisons with the VSV and RSV L structures, the intrusion loop is thought to be displaced as the polymerase complex transitions from pre-initiation to initiation state, allowing repositioning of the priming loop into initiation conformation and accommodation of RNA in the central polymerase cavity [26].

Resistance profiling of ERDRP-0519 and earlier developmental analogs of the chemotype against MeV and CDV has identified specific escape hot-spots in the L RdRP and PRNTase domains [10,12]. A prominent cluster of resistance mutations was located in a sequence-conserved areas immediately before and after the GDNQ catalytic site in RdRP motif C [10,12]. Consistent with the notion that sequence conservation arises from selective pressure on the domain, ERDRP-0519 resistance mutations were associated with viral fitness penalty in cell culture and *in vivo* [10]. The ERDRP-0519 resistance fingerprint is unique among all small-molecule mononegavirus polymerase inhibitors characterized to date, including small molecule inhibitors of closely related respiratory syncytial virus (RSV) such as AZ-27 [32–34] and AVG-233 [35], and a structurally distinct inhibitor of morbillivirus and respirovirus polymerases, GHP-88309, that we have recently described [36]. These features identified the ERDRP-0519 chemotype as a mechanistically first-in-class compound that may have illuminated a conserved, yet currently uncharted, druggable site in mononegavirus polymerases. Despite this high clinical promise, the molecular mechanism of polymerase inhibition by ERDRP-0519 and its physical target docking pose remained unknown, creating a knowledge gap hampering development of the inhibitor class against other mononegavirus targets.

In this study, we have combined biochemical and proteomics strategies to elucidate the mechanistic basis for polymerase inhibition by ERDRP-0519. We have determined the point-of-arrest of polymerase activity through *in vitro* RdRP assays using purified recombinant polymerase complexes and synthetic RNA templates, characterized ERDRP-0519 target affinity and the basis for viral escape through biolayer interferometry, and employed photo-affinity labeling for proteomics-based mapping of the physical target site. Through correlation of these independent datasets, we have developed a docking pose for ERDRP-0519 that defines a novel mechanistic paradigm of mononegavirus polymerase inhibition, pharmacologically locking the polymerase complex in a pre-initiation conformation incapable of initiation RNA synthesis at the promoter. This information allows pharmacophore-guided tuning of the ERDRP-0519 chemotype if called-for by final de-risking prior to clinical development and outlines a blueprint for the structure-informed identification of novel drug candidates directed against mononegavirus polymerases outside of the ERDRP-0519 indication spectrum.

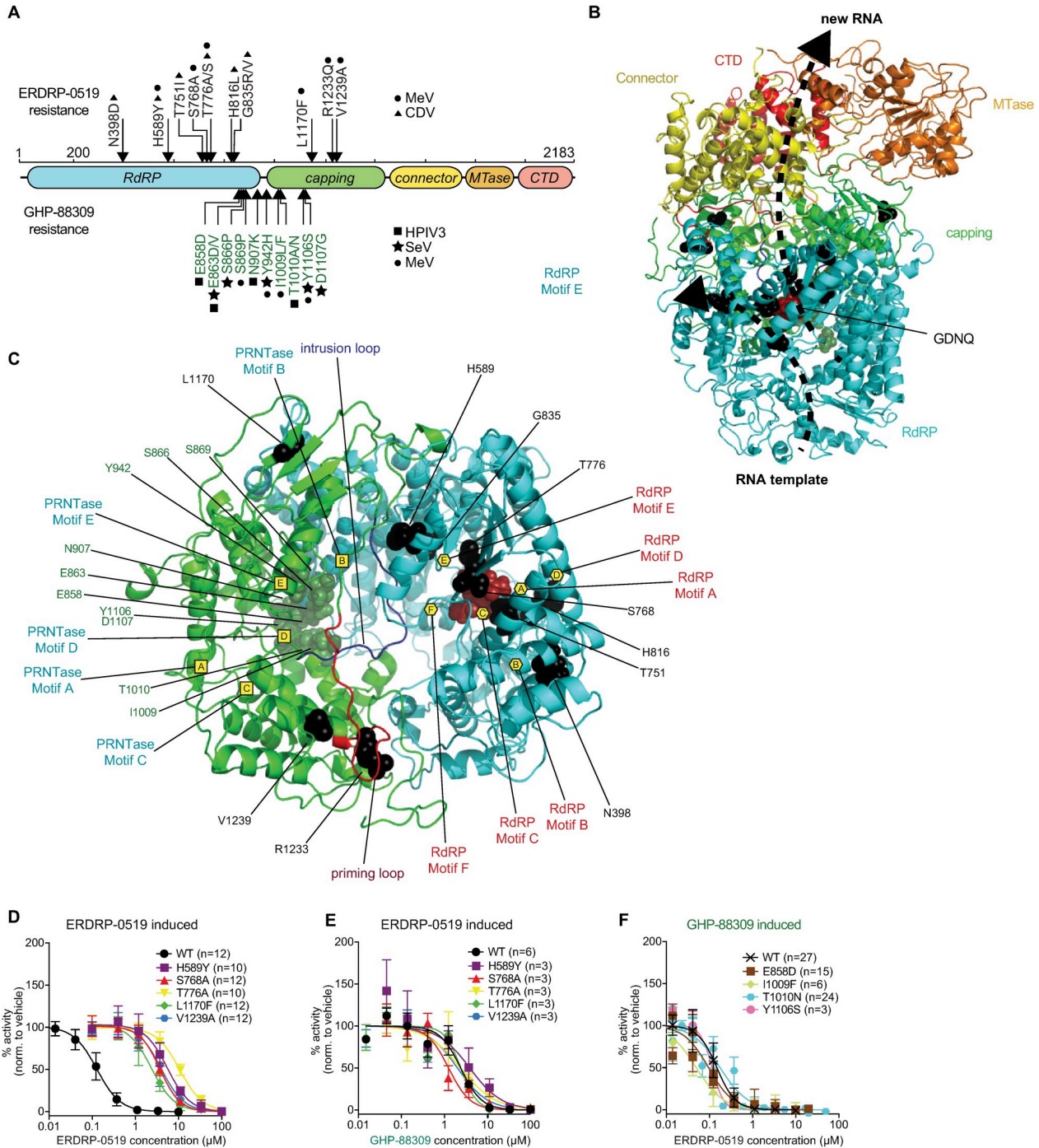

**Fig 1. Resistance profiling of ERDRP-0519 and GHP-88309 against MeV polymerase. A)** 2D-schematic of the MeV L protein. ERDRP-0519 (black) and GHP-88309 (green) resistance mutations are shown. Symbols denote the virus in which each resistance mutation was found (circle, MeV; triangle, CDV; square, HPIV-3; star, Sendai virus). **B)** 3D-homology model of the MeV L polymerase based on the structure of PIV-5 (PDBID: 6v85). The RdRP (cyan), capping (green), connector (yellow), methyltransferase (MTase; orange), and C-terminal (CTD; red) domains are shown. The GDNQ active site is highlighted by red spheres. RNA channels are shown as dotted black lines. **C)** Spatial organization of GHP-88309 and ERDRP-0519 resistance mutations. Shown are locations of tightly clustered GHP-88309 resistance mutations (green) and ERDRP-0519 resistance mutations (black) in the MeV L RdRP (cyan) and capping (green) domains. GDNQ is shown as dark red spheres. Predicted priming and intrusion loops [26] are shown. Proposed PRNTase and RdRP motifs are labeled with yellow squares and hexagons, respectively. The GDNQ active site is highlighted by red spheres. **D-F)** Characterization of resistance mutations in a cell-based MeV minigenome assay. Assessed were ERDRP-0519 induced mutations against ERDRP-0519 (D) and GHP-88309 (E), and GHP-88309 induced mutations against ERDRP-0519 (F). Results for the mutants in (F) against GHP-88309 are summarized in [36]. Symbols show sample means, curves represent 4-parameter variable slope regression models, EC$_{50}$ values are presented in Table 1. The number of biological repeats (n) is specified for each construct.

## Results

We have previously identified signature resistance hot-spots for ERDRP-0519 in MeV and CDV L through viral adaptation [10,12]. Escape sites locate to the polymerase and capping domains of L, but notably do not overlap with the resistance profile of recently developed GHP-88309 [36], the only other well-characterized small molecule inhibitor of MeV L (Fig 1A).

### Pharmacophore of ERDRP-0519 class is unique

To place resistance mutations in a structural context, we highlighted escape sites of ERDRP-0519 and GHP-88309 in a homology model for MeV L based on the PIV-5 L coordinates (Figs 1B and 1C and S1 and Table 1). For further comparison, we also located the MeV L homologs of the known resistance sites to RSV L inhibitors AZ-27, RSV L residue 1631 [34], and an RSV L capping inhibitor series, RSV L residues 1269, 1381, and 1421 [37] (S2 Fig). Sites associated with resistance to ERDRP-0519 broadly line the internal walls of the central L cavity in which template, substrate, and product exit channels converge. AZ-27 and capping inhibitor resistance sites are located distal from the core cavity, forming part of the flexible interface that links the capping and C-terminal methyltransferase domains. With the exception of L residue L1170, all resistance sites were highly conserved across different members of the morbillivirus genus (S3 Fig). GHP-88309 resistance hot-spots are likewise positioned in the core RdRP domain, but these sites were predicted to cluster in the template channel itself rather than the central cavity. Consistent with the notion of non-overlapping docking poses of ERDRP-0519 and GHP-88309 with MeV L, combination treatment of MeV-infected cells revealed a defined zone of synergy of the antiviral activity that was not due to enhanced cytotoxicity (S4 Fig).

To test this hypothesis, we rebuilt ERDRP-0519 and GHP-88309 resistance mutations individually in an MeV minigenome system and examined cross-resistance in dose-response assays. ERDRP-0519-induced mutations reduced MeV polymerase sensitivity to the drug significantly (Table 2), independent of whether they had originally emerged from MeV or CDV adaptation (Fig 1D). However, none of the ERDRP-0519 resistance mutations altered polymerase susceptibility to GHP-88309 (Fig 1E), and none of the previously confirmed substitutions mediating escape from GHP-88309 [36] affected MeV L inhibition by ERDRP-0519 (Fig 1F). The lack of cross-resistance between ERDRP-0519 and GHP-88309 confirms that pharmacophores of these inhibitor classes are distinct.

### Target binding affinity of ERDRP-0519 matches inhibitory concentration

For biochemical positive target identification, we expressed and purified full-length MeV P-L complexes from insect cells (Fig 2A and S1 Data). Mass-spectrometric analysis of the preparation revealed MeV P-L as dominant species in the sample (S1 Table and S2 Data). Prominent among co-purified host proteins included RNA-binding proteins and molecular chaperones. In addition, we prepared a C-terminally truncated MeV L variant ($L_{1708}$) that contains the RdRP and PRNTase domains. This construct remains fully folding-competent as we have demonstrated in previous work [38], but provides a favorable ligand-to-target ratio in biolayer interferometry (BLI). In addition, we purified MeV P-$L_{1708}$ complexes harboring substitutions in one of the three distinct ERDRP-0519 resistance clusters in the linear L sequence, H589Y, S768A or L1170F, respectively, and, for specificity control, RSV-derived P-L complexes. Following *in vitro* mono-biotinylation of the protein preparations, complexes were immobilized on streptavidin high-density BLI probes and ERDRP-0519 association and dissociation curves recorded at increasing compound concentrations (Fig 2B–2D and S5A–S5F). Analysis of ERDRP-0519 binding kinetics to standard full-length MeV P-L and P-$L_{1708}$ complexes

**Table 1. Homology model parameters.** The template PDB ID, virus, sequence identity, sequence similarity, range, coverage, GMQE score and QMEAN for the different homology models used are shown.

| PDB | Virus | Seq Identity | Seq Similarity | Range | Coverage | GMQE | QMEAN |
|------|-------|--------------|----------------|-----------|----------|------|-------|
| 6v85 | PIV5 | 29.72 | 0.35 | 7–2181 | 0.98 | 0.6 | -6.94 |
| 6ueb | RABV | 17.52 | 0.28 | 155–1963 | 0.75 | 0.45 | -6.63 |
| 6uen | RSV | 19.92 | 0.30 | 160–1414 | 0.55 | 0.55 | -5.70 |
| 5a22 | VSV | 17.97 | 0.28 | 197–1961 | 0.70 | 0.31 | -8.41 |

revealed comparable dissociation constants ($K_D$) of 58-140nM (52–59 nM and 78–140 nM for full-length MeV P-L and P-L$_{1708}$, respectively), which resembles the EC$_{50}$ values of 70–230 nM that we had recorded for ERDRP-0519 in cell-based anti-MeV assays [10]. No appreciable binding to RSV polymerase was observed, confirming target protein-specificity of the assay (Fig 2D).

To test whether resistance mutations impair ligand interaction with the L target, we subjected the mutant polymerase complexes to equivalent BLI studies (Fig 2E–2G and S6A–S6F). The mutations increased $K_D$ values to 59 μM (H589Y), 33 μM (S768A), and 48 μM (L1170F), which resembles a 235-fold or greater drop in target affinity. In addition, ERDRP-0519 binding to standard MeV L reached saturation at low micromolar concentrations (S5D–S5F and S7A–S7C Figs) whereas binding to each of the mutant L proteins was drastically reduced and saturation was not reached at the highest concentration tested, 200 μM (S6D–S6F and S7D–S7F Figs). These BLI results positively identify the MeV L protein as ERDRP-0519 target and demonstrate that viral resistance originates from reduced ligand binding affinity.

## ERDRP-0519 resistance mutations restore initiation of RNA synthesis but not RNA elongation

For mechanism-of-action characterization, we applied purified recombinant MeV P-L complexes to *in vitro* RdRP assays in the presence and absence of ERDRP-0519, using $^{32}$P tracers and either a synthetic 16-mer RNA template containing an MeV-specific promoter region to

**Table 2. EC$_{50}$ concentrations against MeV L of resistance mutations induced by viral adaptation to ERDRP-0519 and GHP-88309, respectively.** Binding kinetics ($K_D$) between MeV L$_{1708}$ or MeV L$_{1708}$ harboring resistance mutations and ERDRP-0519 or GHP-88309 are shown.

| L mutation | ERDRP-0519 | | | | GHP-88309 | | | |
|------------|------------|--------------------------------|-------------------|---------|-----------|--------------------------------|-------------------|---------|
| | EC$_{50}$ [μM] | 95% confidence intervals [μM] | fold change to WT | $K_D$ [μM] | EC$_{50}$ [μM] | 95% confidence intervals [μM] | fold change to WT | $K_D$ [μM] |
| WT | 0.13 | 0.11–0.15 | - | 0.079 | 2.40 | 1.94–2.98 | - | 6.2* |
| H589Y[a] | 5.54 | 4.62–6.64 | 44.6 | 59 | 3.91 | 1.87–8.17 | 1.8 | n.d. |
| S768A[a] | 3.72 | 3.35–4.12 | 29.1 | 33 | 1.26 | 0.93–1.71 | 0.6 | n.d. |
| T776A[a] | 10.67 | 9.41–12.10 | 76.6 | n.d. | 2.39 | 1.45–3.94 | 1.1 | n.d. |
| L1170F[a] | 2.36 | 1.99–2.79 | 16.9 | 48 | 2.41 | 1.70–3.4 | 0.7 | n.d. |
| V1239A[a] | 3.97 | 3.34–4.72 | 29.8 | n.d. | 2.03 | 1.34–3.07 | 0.7 | n.d. |
| E858D[b] | 0.10 | 0.08–0.11 | 0.7 | n.d. | 15.04* | 10.69–19.68* | 6.7 | 8.6* |
| I1009F[b] | 0.05 | 0.04–0.07 | 0.3 | n.d. | 18.62* | 13.78–25.51* | 8.3 | 68.3* |
| T1010N[b] | 0.18 | 0.15–0.21 | 1.5 | n.d. | 8.36* | 4.40–15.71* | 3.8 | n.d. |
| Y1106S[b] | 0.11 | 0.09–0.14 | 0.6 | n.d. | >100* | >100* | >50 | >300* |

* reported previously [36].

a resistance mutation induced by ERDRP-0519.

b resistance mutation induced by GHP-88309.

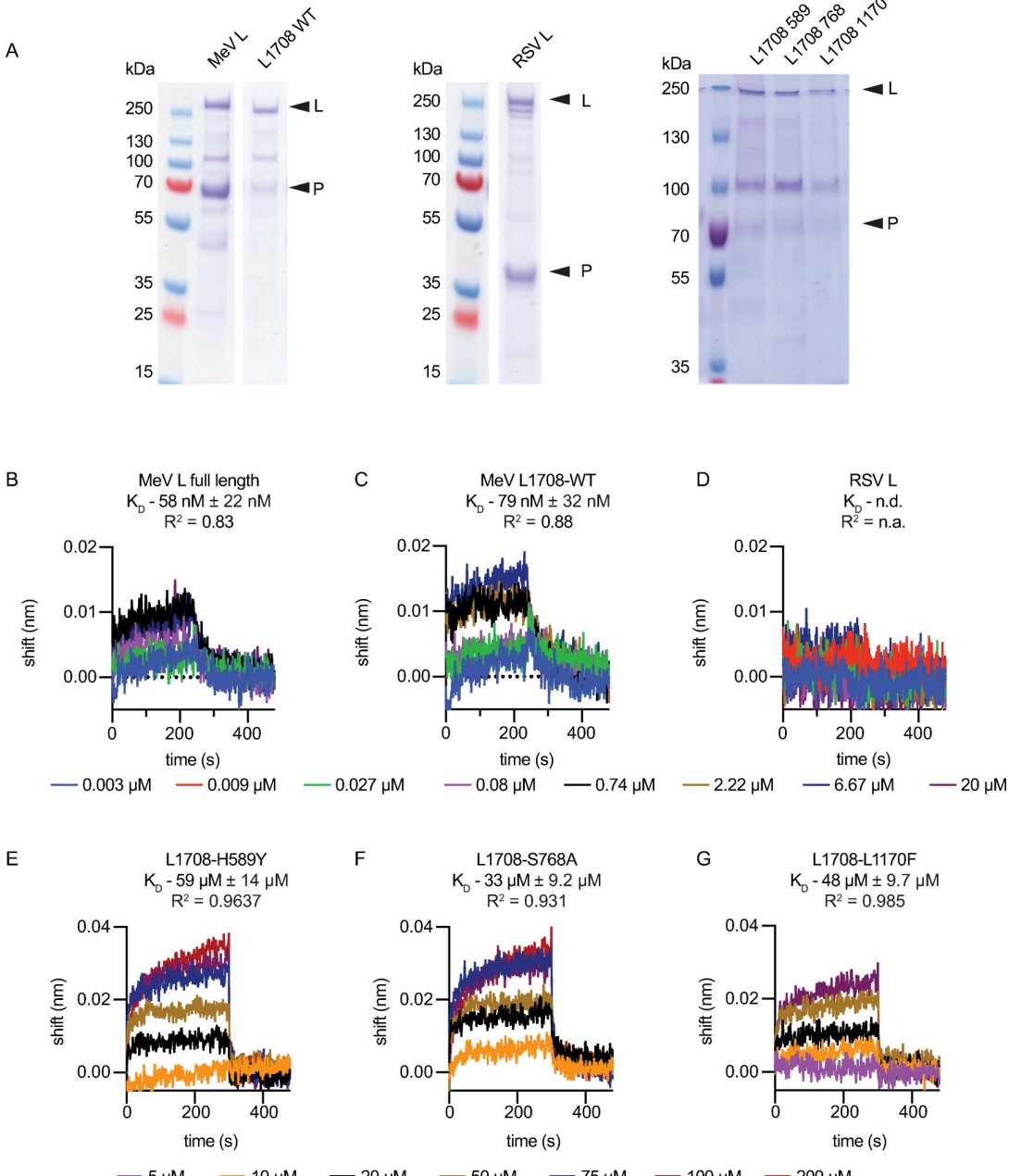

**Fig 2. Target binding affinity of ERDRP-0519. A)** SDS-PAGE of purified full-length MeV L, MeV L$_{1708}$ [36], and RSV L used for BLI studies. **B-G)** BLI of ERDRP-0519 and purified standard (WT) MeV L (B), MeV L$_{1708}$ (C), RSV L (D) and MeV L$_{1708}$ harboring selected ERDRP-0519 resistance mutations (E-G). Similar binding kinetics were observed for full-length MeV L and MeV L$_{1708}$. K$_D$ values and goodness of fit are shown for each construct.

monitor *de novo* initiation of RNA synthesis at the promoter (Fig 3A) or a synthetic primer-template pair to assess RNA elongation by a committed polymerase (Fig 3B). Autoradiograms and phosphoimager-based quantitations revealed efficient and dose-dependent inhibition of standard MeV polymerase complexes under either assay condition with IC$_{50}$ values of 0.15 μM and 0.1 μM, respectively (Fig 3C–3F and S3 and S4 Data). These active concentrations closely recapitulated the EC$_{50}$ of ERDRP-0519 against MeV in cell-based assays. Catalytically defective

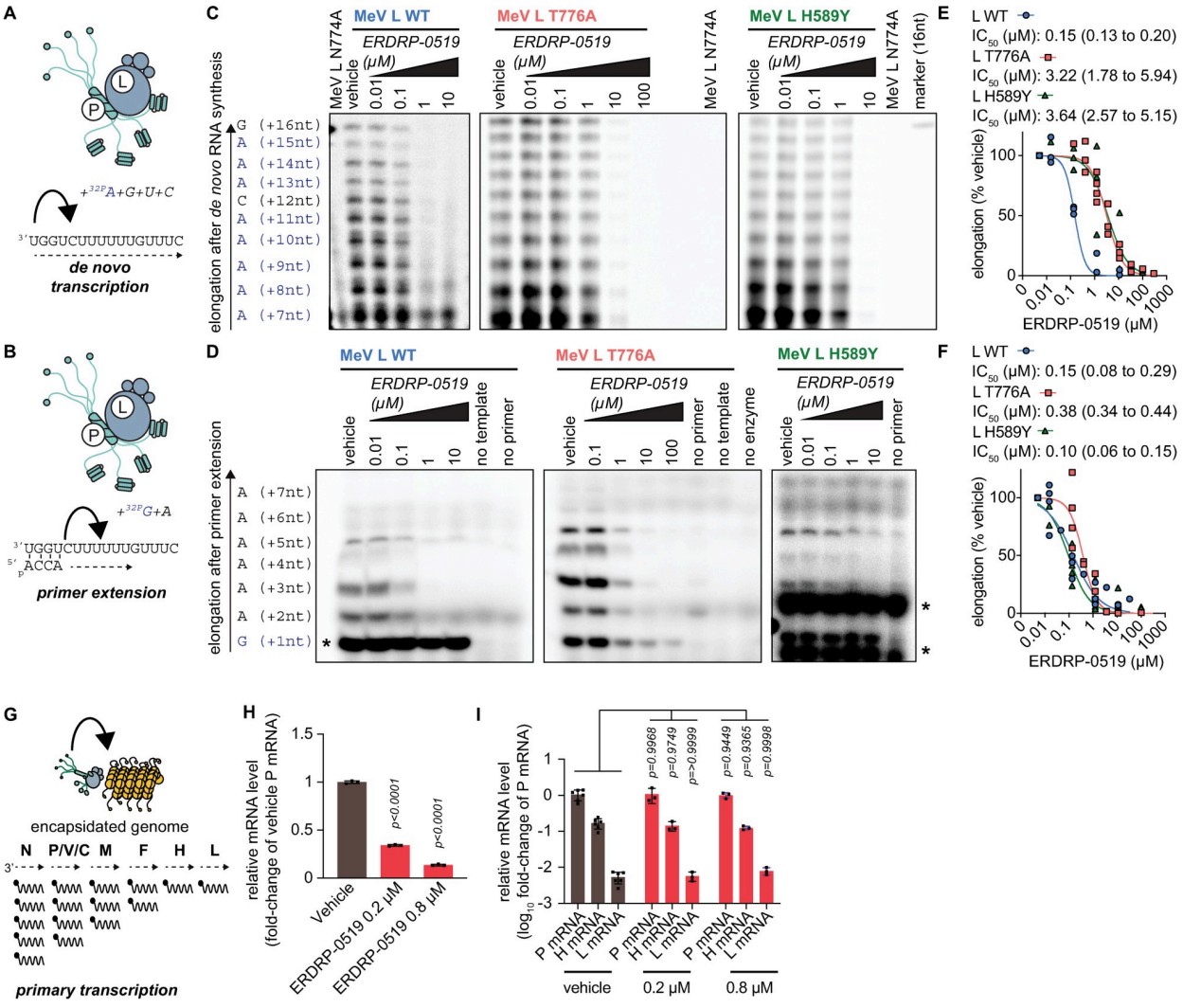

**Fig 3. ERDRP-0519 potently inhibits *de novo* RNA synthesis. A-F)** Purified recombinant WT MeV L-P complexes or complexes harboring the L$_{T776A}$ or L$_{H589Y}$ resistance mutations were incubated with either a 16-nt RNA template (A) and the represented NTPs to assess *de novo* RNA synthesis or a 16-nt RNA template (B), a 5'-phosphorylated 4-nt primer, and the represented NTPs to assess primer extension. Representative autoradiograms are shown for *de novo* initiation (C) and primer extension (D). Purified L-P complexes with an L$_{N774A}$ substitution in the catalytic GDNQ motif [70] served as control for specificity of the *de novo* initiation assay. Primer extension assays were performed in the absence of primer, template, or enzyme to control for contaminating *de novo* RNA synthesis driven directly by the template. Densitometry analysis was performed on elongation products of 15 to 16-nt in length (E) or 7 to 9-nt in lengths (F) and represent n = 3–4 biological repeats. EC$_{50}$ values represent 4-parameter variable slope regression models, 95% confidence intervals are shown. Positions of unspecific background signals are indicated with *.
**G-I)** Effect of ERDRP-0519 incubation on MeV primary transcription, represented in (G). Cells were infected with recMeV-Anc (MOI = 3) and incubated with vehicle (0.1% DMSO) volume control, or 0.2 μM or 0.8 μM ERDRP-0519. Infected cells were harvested 4 hours after infection. Incubation with ERDRP-0519 significantly decreased relative P-encoding mRNA amounts 4 hours post infection (H), but did not alter steepness of the MeV primary mRNA transcription gradient (I). Symbols represent individual biological repeats (n = 3), graphs show sample means. Statistical analysis through two-way ANOVA with Dunnett's multiple comparison post-hoc test, P values are specified.

L mutants harboring an N774A substitution (Fig 3C) or a D773A N774A double-mutation (S8 Fig and S5 Data) in the polymerase active site confirmed that RNA synthesis was MeV P-L specific and not due to co-purified cellular contaminants.

Polymerase complexes containing an ERDRP-0519 resistance mutations H589Y or T776A showed greatly reduced susceptibility to compound-mediated suppression of *de novo* polymerase initiation, reflected by inhibitory concentrations of 3.64 μM and 3.22 μM, respectively,

which represents a 24- and 21-fold increase in $IC_{50}$ value (Fig 3E). By comparison, the same resistance mutations had a marginal effect on lifting ERDRP-0519 inhibition of RNA elongation in the *in vitro* RdRP assay. The H589Y substitution did not increase the $IC_{50}$ concentration at all ($IC_{50}$ 0.10 μM) and T776A caused only a moderate 2.5-fold $IC_{50}$ increase ($IC_{50}$ 0.38 μM) (Fig 3F). Because these resistance mutations mediate robust viral escape in cell-based inhibition assays, the *in vitro* RdRP results highlight inhibition of RNA synthesis at the promoter as the physiologically relevant antiviral effect of ERDRP-0519. Resistance mutation-insensitive inhibition of primer extension is very likely a consequence of the artificial conditions of the *in vitro* assay using a non-encapsidated template that does not fully recapitulate the natural system.

To test this hypothesis, we assessed the effect of ERDRP-0519 on the relative amount of viral mRNA transcripts synthesized in infected cells during the first 4 hours after infection, representing the primary transcription phase (Fig 3G). MeV P mRNA levels were dose-dependently reduced by ERDRP-0519 (Fig 3H). Importantly, however, the inhibitor had no significant effect on the ratio of different viral mRNAs relative to each other (Fig 3I), the viral transcription gradient. This finding indicates that ERDRP-0519 is unable to inhibit RNA elongation by an engaged polymerase complex in infected cells, which would steepen the transcription gradient.

## ERDRP-0519 blocks all phosphodiester bond formation

All allosteric paramyxovirus and pneumovirus polymerase inhibitors subjected to *in vitro* RdRP assays to date were found unable to block the formation of the first phosphodiester bond during backpriming and/or primer extension [35,36,39]. Backpriming refers to the spontaneous formation of a circular hairpin structure of the non-encapsidated synthetic templates [40] that allows extension of the resulting paired 3'-ends beyond the actual length of the template (Fig 4A). To assess whether the same limitation applies to ERDRP-0519, we applied the MeV polymerase complex to a previously described 25-mer RNA template derived from an authentic RSV promoter sequence that is capable of efficient backpriming [40]. The RSV template was efficiently recognized as a suitable substrate for *de novo* RNA synthesis by the MeV polymerase (Fig 4B and S6 Data). Potency of inhibition of *de novo* initiation by ERDRP-0519 resembled that observed for the MeV promoter ($IC_{50}$ 0.12 vs 0.15 μM). In contrast to all available inhibitors formerly described, however, all 3'-elongation after backpriming was equivalently dose-dependently blocked (Fig 4C).

For validation of this unprecedented finding, we carried out a primer extension assay in the absence of all NTPs but the $^{32}P$-GTP tracer, consequently allowing monitoring incorporation of the first nucleotide, specifically (Fig 4D and S7 Data). Again, ERDRP-0519 dose-dependently blocked incorporation of the very first nucleotide with equivalent potency for standard ($L_{WT}$; $IC_{50}$ 0.12 vs 0.15 μM) and resistant ($L_{T776A}$; $IC_{50}$ 0.3 vs 0.39 μM) MeV polymerases to inhibition of multi-nucleotide elongation (Fig 4B).

In conclusion, these results consistently demonstrate that ERDRP-0519 directly suppresses all phosphodiester bond formation, setting the compound mechanistically apart from all other allosteric mononegavirus inhibitor classes characterized to date.

## Photoaffinity labeling maps the ERDRP-0519 target site to the central L cavity

To map the physical target site of ERDRP-0519, we synthesized a photo-activatable analog of the compound, ERDRP-0519$_{az}$, through installation of an aryl azide moiety at C-2 position of the ERDRP-0519 piperidine ring via a short tether (Fig 5A). Analog design was guided by our

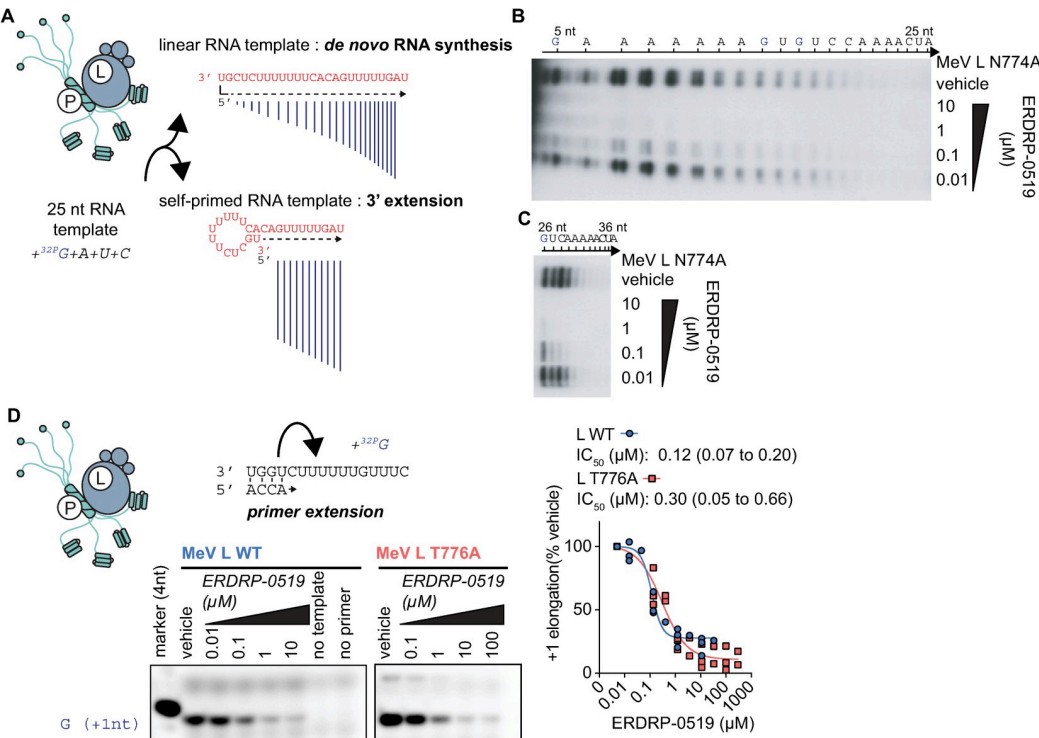

**Fig 4. Effects of ERDRP-0519 on single nucleotide addition. A-C)** Purified recombinant P-L complexes were incubated with the specified NTPs and a 25-nt RNA template driving both *de novo* initiation and back-priming [40] (A). A representative autoradiogram (n = 3) is shown, divided for clarity into dose-dependent inhibition of *de novo* RNA synthesis initiation at the promoter by ERDRP-0519 (B) and inhibition of 3'-elongation after back-priming (C). Products of less than 5-nt are not perceptible due to background from unincorporated $^{32}$P-labelled nucleotides. **D)** Dose-dependent inhibition of primer extension by ERDRP-0519 as in Fig 3B, but only GTP was added to visualize incorporation of the first nucleotide. $EC_{50}$ values represent 4-parameter variable slope regression models, 95% confidence intervals are shown (n = 3).

extensive insight into the structure-activity relationship (SAR) of the ERDRP-0519 chemotype that we had acquired in previous work [11]. Bioactivity testing of ERDRP-0519$_{az}$ against MeV revealed dose-dependent suppression of virus replication and an $EC_{50}$ of 12.1 μM (Fig 5B). No cytotoxicity was detectable at the highest concentration tested (100 μM). Compared to ERDRP-0519, antiviral potency of ERDRP-0519$_{az}$ was reduced approximately 50-fold. Assessment of ERDRP-0519$_{az}$ in the *in vitro* RdRP assay confirmed bioactivity of the analog with diminished potency (S9 Fig and S8 Data), suggesting a slightly impaired ability to engage the target. Since both assays confirmed specific inhibition of the MeV polymerase by the photo-activatable probe, however, we compensated for this potency reduction by incubating the P-L complexes in the presence of 40 μM compound prior to photo-activation.

Photo-coupling of ERDRP-0519$_{az}$ to purified L followed by LC-MS/MS analysis after trypsin digestion of the ligand-L complexes identified three discrete peptides that are located in the capping and RdRP domains of the L protein (Fig 5C). Peptides 1 and 2 (spanning residues 952–966 and 1151–1167, respectively) are located in immediate proximity to each other, forming part of the upper wall of the central polymerase cavity (Figs 5D and S10). Peptide 3 spans residues 630–666 in the RdRP domain, a poorly structured region in mononegavirus polymerases. Considering the high structural variability of this sequence stretch and simultaneous covalent binding of two ERDRP-0519$_{az}$ moieties to peptide 3, this peptide emerged as a very likely candidate for nonspecific crosslinking.

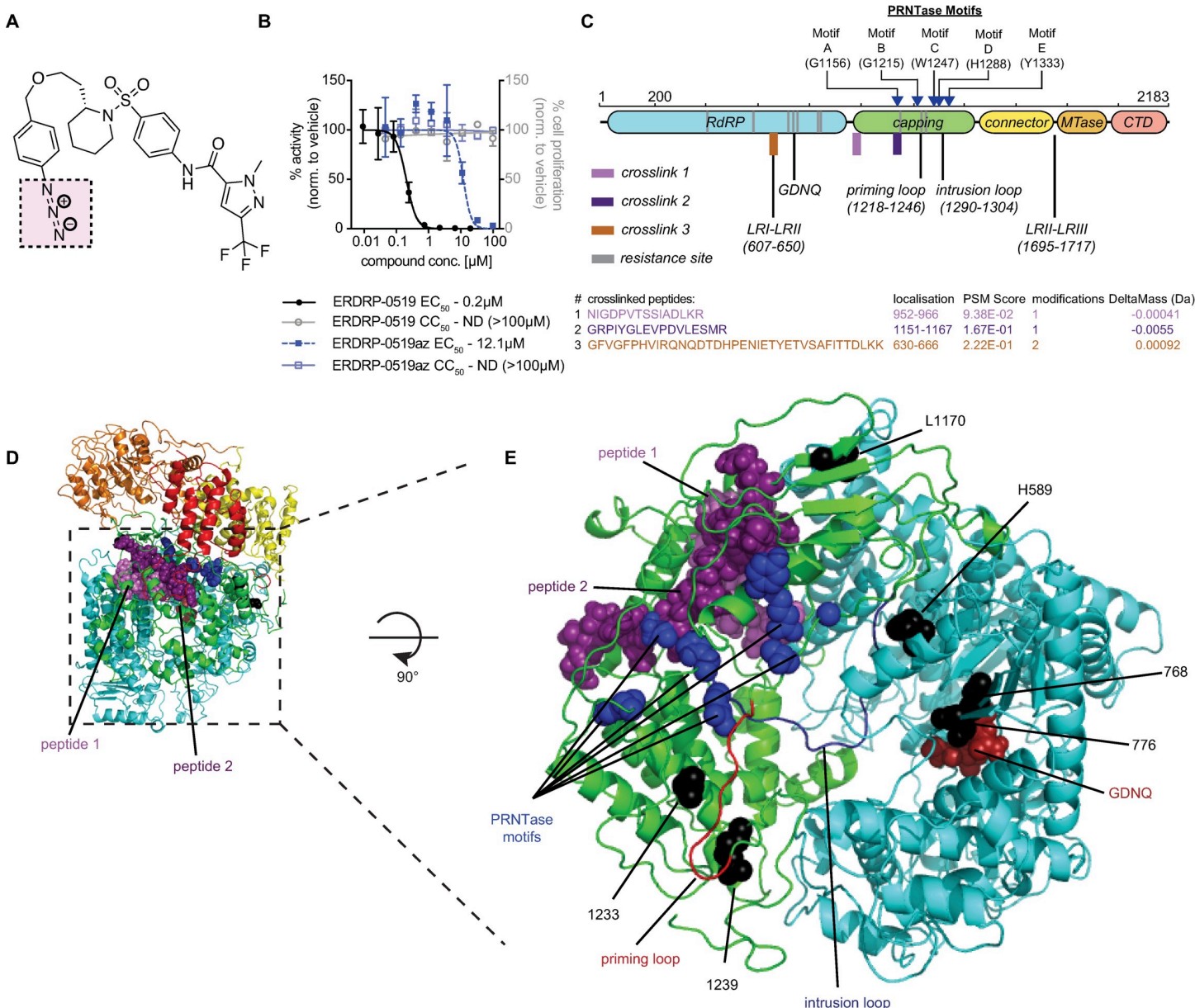

**Fig 5. Photoaffinity labeling-based target mapping of ERDRP-0519. A)** Structure of photoactivatable compound ERDRP-0519$_{az}$; the reactive azide moiety is highlighted (pink square). **B)** ERDRP-0519$_{az}$ is bioactive and displays no appreciable cytotoxicity. Symbols show means of three biological repeats, graphs represent 4-parameter variable slope regression models where possible. EC$_{50}$ and CC$_{50}$ are shown. **C)** 2D-schematic of the MeV L protein with locations of crosslinked peptides identified by photoaffinity labeling (top). The RdRP (cyan), capping (green), connector (yellow), MTase (orange), and C-terminal (light-red) domains, locations of known unstructured regions (LRI-LRII; LRII-LR-III), the GDNQ active site, specific amino acids motif in the capping domain, and positions of the intrusion and priming loops are shown. Sequences of peptides engaged by ERDRP-0519$_{az}$ (bottom). Specified are peptide location, spectrum match (PSM) score, number of ligands present, and delta mass. **D)** Location of peptides 1 (pink) and 2 (purple) in MeV L. **E)** Close-up top view of the capping and RdRP domains showing the adjacent locations of peptides 1 and 2. The priming loop (red), intrusion loop (blue), PRNTase motifs (blue spheres) and GDNQ active site (red spheres) are shown.

We therefore concentrated further analysis on peptides 1 and 2, which are near the intersection between the polymerase capping, connector and MTase domains, and in close proximity to highly conserved polymerase motifs such as the HR moiety of motif D and G1156 of motif A of the PRNTase domains as well as both the postulated paramyxovirus L priming and intrusion loops [26] (Fig 5E).

## Docking predicts that ERDRP-0519 locks the polymerase in pre-initiation conformation

Overlaying the photo-crosslinking results and resistance maps in the MeV L structural model revealed a circular arrangement of all potential ERDRP-0519 anchor points along the interior lining of the central polymerase cavity (Fig 6A). Docking of ERDRP-0519 into the L structure was guided by the following constraints: positioning of the ligand is compatible with the formation of covalent bonds with residues in photo-crosslinking defined peptides 1 and 2; and the docked compound is in equal proximity to confirmed resistance sites H589, S768, T776, L1170, R1233, and V1239. In addition to ERDRP-0519, two chemical analogs, the original screening hit 16677 [13] and the first-generation lead AS-136 [41], were used as ligands for *in silico* docking (S11 Fig). A conserved top-scoring pose for 16677, AS-136, and ERDRP-0519 was returned that placed each analog in a pocket at the intersection of MeV L capping and RdRP domains, between PRNTase motifs A and D (Fig 6A). This pose positions the ligand in approximately 26Å distance to peptide 1 and 2.7–10.4Å distance to peptide 2, establishing hydrogen bond interactions between residue Y1155 in peptide 2 and the central pyrazole ring of the ERDRP-519 scaffold (Fig 6B). Additional hydrogen bonds are predicted between H1288 in the polymerase HR motif [26] and the sulfonyl group of the ERDRP-0519 scaffold, and N1285 of the intrusion loop and the piperidine moiety of ERDRP-0519. Sequence alignments revealed that core residues predicted to be in close proximity to the bound ligand are highly conserved in the morbillivirus genus (S12 Fig). Equivalent ERDRP-0519 docking poses could not be identified for PIV-5, RSV, and VSV L (S13 Fig) which are all insensitive to the compound.

In top-scoring position, ERDRP-0519 should lock MeV L intrusion and postulated priming loops in place, blocking reorganization of the central cavity for polymerase initiation. To test this hypothesis, we attempted to dock the ligand into the equivalent space in an MeV L model based on the structure of VSV L [24], in which the priming loop is considered to be in initiation conformation, displacing the extended intrusion loop of the PIV-5 L structure from the central polymerase cavity. Only in this configuration can the polymerase accommodate RNA in the cavity [42]. However, the reorganization of the PRNTase domain into initiation mode eliminated the ERDRP-0519 binding pocket (Fig 6C and 6D), suggesting that the compound stabilizes a pre-initiation conformation of the polymerase.

To explore whether the docking pose supports covalently binding of photo-reactive ERDRP-0519 to experimentally identified target residues, we examined docking of ERDRP-0519$_{az}$ in the pre-initiation L model. A top-scoring pose resembled that of standard ERDRP-0519 (Fig 6E and 6F). However, the terminal ring system carrying the azide arm was rotated by 180˚ (Fig 6G) and was predicted to covalently engage L1157 in peptide 2. This pose supports peptide 2 as a premier covalent target site for ERDRP-0519$_{az}$. However, ERDRP-0519$_{az}$ did not hydrogen bond with Y1155 and H1288, which may explain the approximately 60-fold decrease in potency of ERDRP-0519$_{az}$ compared to standard ERDRP-0519.

## Ligand-driven 3D-quantitative SAR validates the *in silico* docking-derived pharmacophore

To independently assess the predictive power of the photoaffinity labeling-based docking pose, we selected from our synthetic collection an informative set of 33 analogs of the ERDRP-0519 chemotype with EC$_{50}$ values ranging from 0.005–65 μM (S2 Table) [11,41] to generate a ligand-driven 3D-QSAR model. This panel was divided into a training subset of 16 analogs with an EC$_{50}$ range of 0.005 to 23 μM and a 17-member test subset. We next generated an *in silico* library of 25–523 distinct conformations for each compound of the training set and

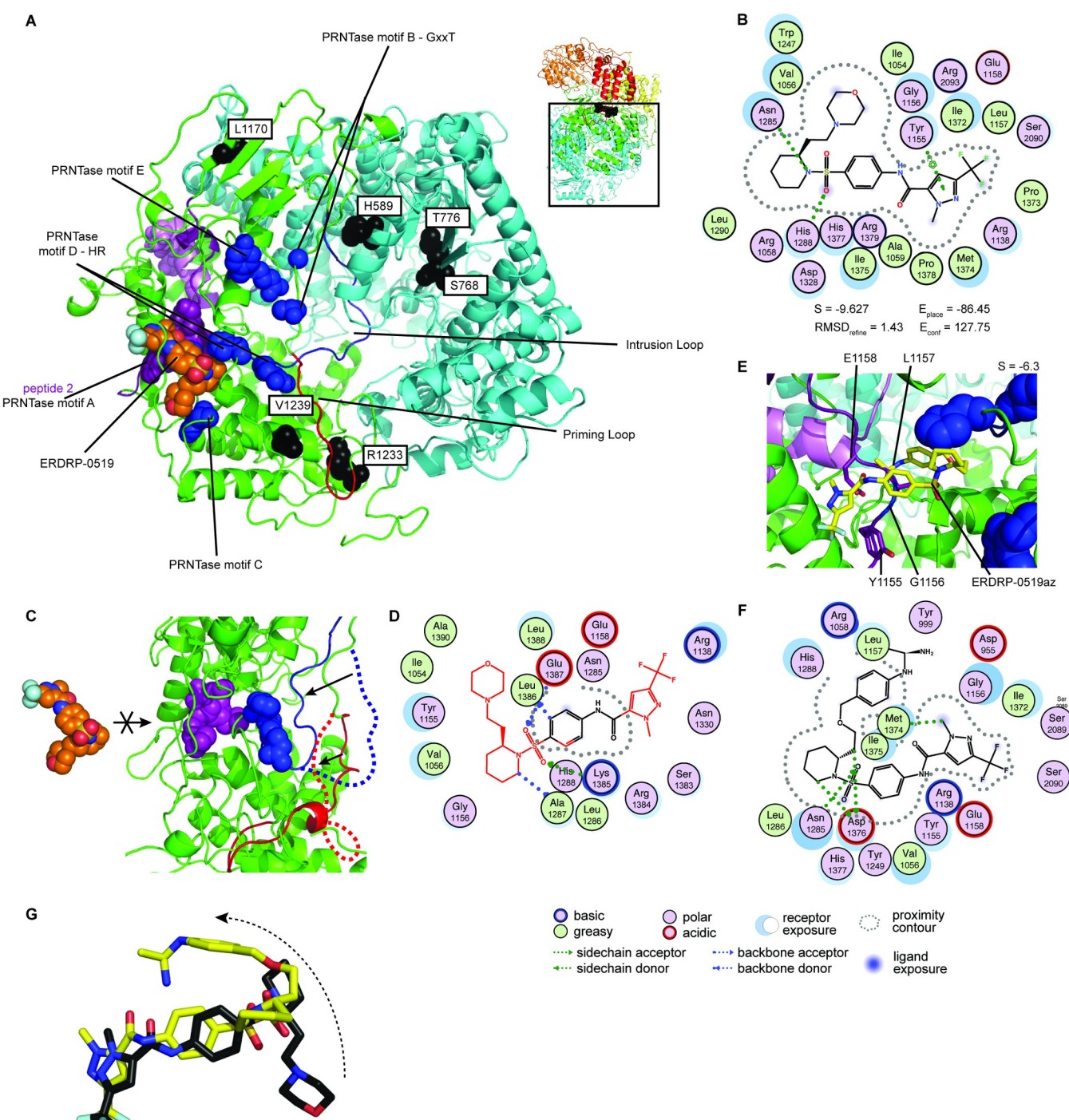

**Fig 6. *In silico* docking and ERDRP-0519 pharmacophore extraction. A)** Top view of the RdRP (cyan) and capping (green) domains of MeV L. PRNTase motifs, ERDRP-0519 resistance mutations, and ERDRP-519 are shown as blue, black, and magenta spheres, respectively. Peptide 2 (purple), the priming loop (red) and intrusion loop (blue) are labeled. The top scoring docking pose places ERDRP-0519 (orange) in close proximity to peptide 2. **B)** 2D-interaction projection of the top scoring ERDRP-0519 docking pose. The sulfonyl oxygen is predicted to interact with H1288 of the HR motif and the pyrazol ring hydrogen bonds with Y1155 of peptide 2. **C-D)** The binding pocket of ERDRP-0519 is not available in an MeV L homology model based on VSV L in initiation conformation (PDBID: 5A22) (C). Attempts to insert ERDRP-0519 result in multiple steric violations (moieties of the compound structure highlighted in red) (D). **E-G)** Close-up view (E), 2D-interaction map (F), and scaffold overlays (G) of the corresponding top-scoring covalent ERDRP-0519az docking pose. Positioning and main scaffold orientation resembles the pose of ERDRP-0519, although the sulfonyl group hydrogen bonds with D1378 instead of H1288. The lateral ring system containing the azide moiety of ERDRP-0519az (yellow sticks) must rotate (G) to fit.

aligned these into 287 initial pharmacophore models that were generated with the AutoGPA module embedded in the MOE software package [43]. Individual models were ranked based on goodness of fit ($R^2$) and predictive capacity ($q^2$) against the training set (Fig 7A), leading to the identification of a top-scoring 3D-QSAR model with $R^2 = 0.9749$ and $q^2 = 0.7768$. Predicted and experimentally determined activity of the training subset were correlated with a regression slope of 0.9803 through the origin. Validation of this model against the 17-analog test subset that included ERDRP-0519, ERDRP-0519$_{az}$, and the first generation lead AS-136 returned an $R^2$ of 0.54 and regression slope of 0.89 for the test set of analogs (Fig 7A).

Direct comparison of the 3D-QSAR-derived pharmacophore and the top-scoring docking pose revealed a close overlap of the predicted conformation of the target-bound ERDRP-0519 core scaffold (root-mean-square-deviation (RMSD): 5.484; Fig 7B). Most notable of the highly conserved features between both pharmacophores are the predicted strong hydrogen bond interactions between the target pocket and the piperidine and sulfonyl groups of the ERDRP-0519 scaffold (Fig 7C). Graphical overlay of both models furthermore revealed a strong correlation between shape and dimension of the available space in the target site illuminated by the L protein structural model and the ligand-occupied space requested by the 3D-QSAR model (Fig 7D–7F).

## Discussion

Despite the major clinical threat posed by mononegaviruses and several decades of drug development, ribavirin is the only small-molecule antiviral therapeutic licensed for clinical use against any pathogen of the order to date. However, ribavirin for treatment of RSV infections has been largely discontinued due to a poly-pharmacological mechanism of action [44–46], pronounced side effects [47,48], and limited efficacy [49], creating a major unaddressed clinical need. We favor the mononegavirus RdRP complex as a premier druggable target, based on its diverse enzymatic activities, its unique catalytic activity that lacks a cellular equivalent, and its critical importance for both viral replication and the expression of non-structural immunomodulatory viral proteins that counteract the host innate antiviral response [50]. Groundbreaking progress in the structural understanding of the organization of mononegavirus polymerase complexes [23–26] has established a foundation to define distinct druggable sites in the L polymerase.

Focusing on the closely related pneumovirus and paramyxovirus families specifically, four distinct non-nucleoside chemotypes have been subjected to defined RdRP assays using purified polymerase complexes and synthetic RNA substrates. Of these, two (AZ-27 [39] and AVG-233 [35]) are RSV L-specific, one (GHP-88309 [36]) inhibits paramyxoviruses of the respiro- and morbillivirus genera, and ERDRP-0519 specifically blocks morbillivirus polymerases [10,12]. All of these compounds have been demonstrated to interfere with *de novo* polymerase initiation at the promoter and backpriming, albeit in the case of AZ-27, AVG-233, and GHP-88309, after incorporation of an additional 2–4 nucleotides [35,36,39]. This delayed polymerase arrest demonstrates that these compounds do not directly block phosphodiester bond formation. Structural evidence [23–26,51] and functional characterization [35,39] rather indicates pharmacological interference with conformational changes of the polymerase as the enzyme transitions from initiation to RNA elongation mode. Consistent with this conclusion, none of these three inhibitor classes affects extension of the RNA template after backpriming [35,36,39], which mimics RNA elongation by a committed polymerase complex [40,52,53].

Characterization of ERDRP-0519 demonstrated that preventing the switch to elongation mode is not the only mechanism available to allosteric small molecule inhibitors, since the compound interrupted both initiation at the promoter and RNA elongation after backpriming

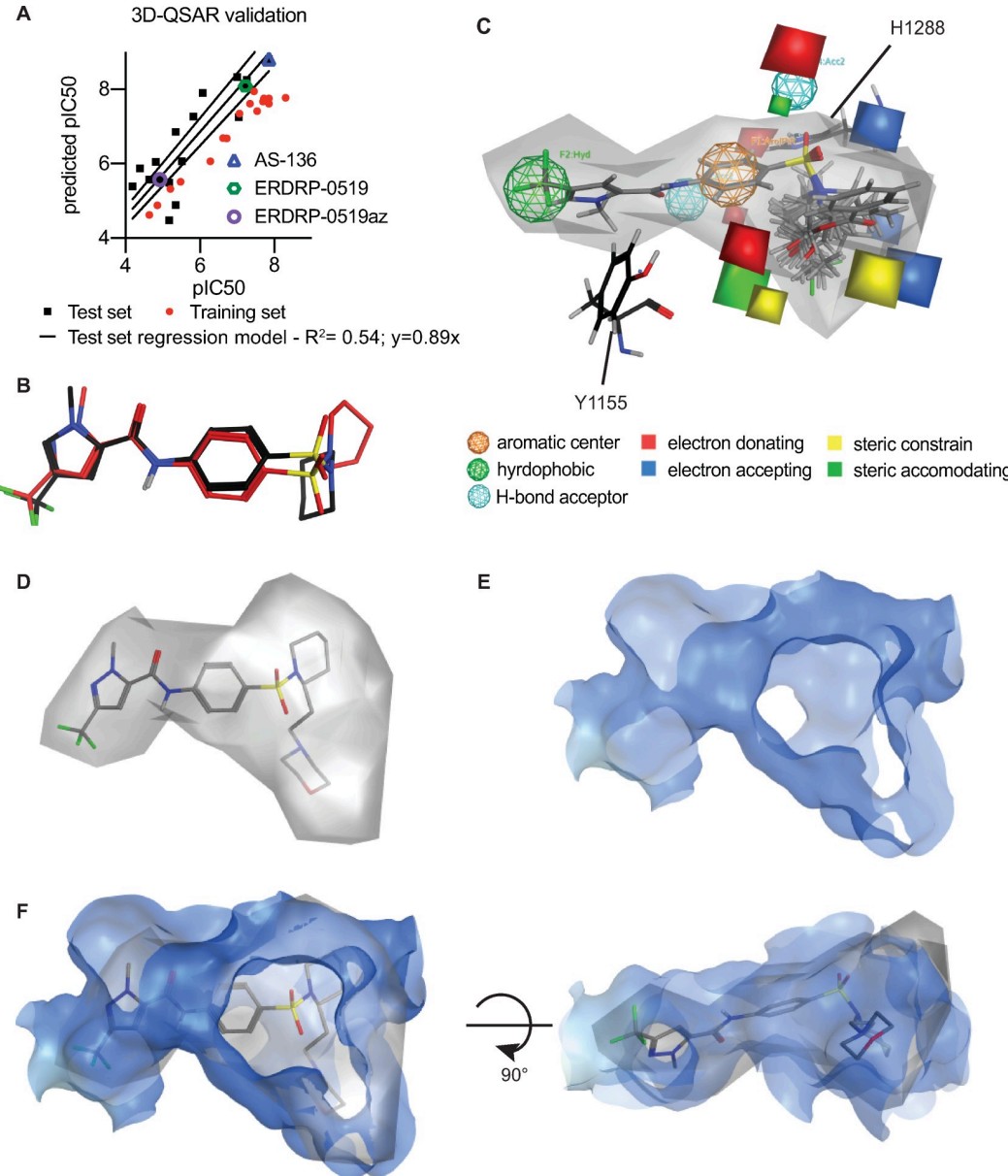

**Fig 7. Independent ERDRP-0519 docking validation through ligand-driven 3D-QSAR modeling. A)** Development of the 3D-QSAR model. Individual data points of the training and test set, goodness of fit ($R^2$), and slope of the best fit correlation through the origin are shown for the test set. **B)** Overlays of docking poses of the core ERDRP-0519 scaffold identified by the 3D-QSAR (red) and photocrosslinking-informed *in silico* docking (black). **C)** Graphical representation of the 3D-QSAR model showing space filling (grey) and pharmacophore features of the model. Proximity of features to H1288) and Y1155 is shown. **D-F)** Overlays of the space filling portion of the 3D-QSAR model (grey) with the L-provided binding pocket (blue). Individual views of the space-filling portion (D) and the binding pocket (E), and top and side views of the overlays (F) are shown.

with equal potency. Underscoring a unique mechanism of action, inhibition of polymerase initiation was immediate, indicating that all phosphodiester bond formation ceases in the presence of the inhibitor. As a consequence of this MOA principle, ERDRP-0519 potency in the RdRP closely resembled that in cell based antiviral assays. This behavior could not be taken for granted, since other polymerase inhibitor classes showed a remarkable discrepancy between *in*

*vitro* and *in cellula* active concentrations [35,36,39]. Only target binding affinity of ERDRP-0519 determined through BLI furthermore resembled the inhibitory concentration range, indicating that in the case of ERDRP-0519 interaction with purified P-L complexes assesses compound docking into the primary, bioactivity-relevant target site. This finding provided support also for the physiological relevance of the photoaffinity labeling-based target mapping strategy that likewise relied on *in vitro* interaction of the ligand with purified polymerase complexes.

Although three peptides were identified by photocrosslinking, we primarily focused on peptides 1 and 2, since peptide 3 is located in a highly variable and unstructured region that is solvent exposed in the L structural model (S10 Fig). Two ERDRP-0519az molecules bound furthermore simultaneously to the same peptide 3 molecule, underscoring nonspecific crosslinking. Peptides 1 and 2 are separated by some 200 amino acids in the linear polypeptide sequence, but are located in close proximity to each other in the native PRNTase domain. Physical engagement of residues in either of these two peptides by ERDRP-0519az in a consistent, specific docking pose was structurally only possible if the compound is positioned in the central cavity of the polymerase complex.

The top-scoring docking pose proximal to peptide 2 offers a mechanistic hypothesis for how ERDRP-0519 can prevent all phosphodiester bond formation. The lead candidate residues in peptide 2 for covalent interaction with ERDRP-0519az, Y1155, G1156, L1157, and E1158, are all located at a critical intersection between the polymerase capping and RdRP domains. Direct interaction between ERDRP-0519 and the PRNTase intrusion and priming loops sterically blocks repositioning of the intrusion loop towards the cavity wall as seen in the VSV, RABV, and RSV L structures (Figs 6C and S14) [23–25] that is considered essential for polymerase initiation. The extended intrusion loop blocks access of template RNA to the cavity [26], and prevents rearrangement of the priming loop into initiation conformation. Such pharmacologically locked into a pre-initiation conformation that cannot accommodate template RNA in the central cavity, the polymerase complex is unable to initiate +1 RNA synthesis or elongate RNA in primer extension and after backpriming. In addition, H1288 of the HR moiety in PRNTase motif D is essential for the initiation of L-mediated RNA synthesis [54]. Hydrogen bonding between the sulfonyl group of ERDRP-0519 and this residue is very likely catastrophic for the formation of productive initiation complexes.

Confirmed ERDRP-0519 resistance sites line the internal wall of the central polymerase cavity, but individual hot-spots poorly cluster in the native structure and none is predicted to be in direct contact with the docked ligand. Remarkably, however, resistance sites were located in highly sequence conserved L domains such as on the priming and intrusion loops (i.e. H589Y, R1233Q, and V1239A) and/or in immediate proximity of functional motifs (i.e. S768A and T776A framing the GDNQ catalytic center). We conclude that escape from ERDRP-0519 is mediated by secondary structural effects rather than due to primary resistance, which is unusual for non-nucleoside analog polymerase inhibitors [55–57]. Conceivably, these substitutions could shift the spatial preference of intrusion and priming loops in the pre-initiation polymerase complex [26], favoring orientation of the intrusion loop towards the cavity wall that is incompatible with compound binding. Naturally, conformational flexibility of the loops cannot be eliminated entirely without loss of polymerase bioactivity. Three observations support the model that resistance arises from a shift in conformational equilibrium of the loops: i) the maximally achievable degree of resistance is moderate and escape can be readily overcome through increased compound concentrations; ii) the loss in compound bioactivity due to resistance resembles the drop in target binding affinity; and, iii) escape from ERDRP-0519 in all cases carries a substantial viral fitness penalty [10], consistent with disturbed, but not obliterated, polymerase function.

Most notably, resistance mutations greatly eased the ERDRP-0519 block of *de novo* initiation of RNA synthesis in *in vitro* RdRP assays, but had only a marginal effect on restoring RNA elongation in primer extension. This phenotype reveals that restoring RNA synthesis initiation in the presence of compound constitutes the dominant selective force for resistance, identifying inhibition of *de novo* polymerase initiation at the promoter as the primary antiviral effect of ERDRP-0519 in the infected cell. The predicted docking pose of ERDRP-0519 in the template RNA channel is consistent with the notion that under physiological conditions the compound cannot engage and arrest an already committed polymerase complex in RNA elongation mode. Pharmacological interference with RNA elongation in the *in vitro* RdRP assay likely represents an artifact caused by the short and non-encapsidated synthetic template RNA that does not fully recapitulate physiological conditions. Further experimental support for this conclusion comes from the unchanged steepness of the viral mRNA transcription gradient in infected cells exposed to ERDRP-0519, confirming that the polymerase complex is not susceptible to inhibition any longer once RNA elongation has started.

The ERDRP-0519 docking pose into MeV L provides a direct molecular explanation for the morbillivirus-specificity of the compound, since L microdomains forming the physical binding site of the inhibitor show considerable variability between different genera in the paramyxovirus family (S1 Fig) [10,12]. To date, native structural information for paramyxovirus polymerases is available only for PIV-5 L, and the resolution of the structure prevents atomic localization of individual side chains [26]. Although these restrictions make structure-guided ligand design to broaden the indication spectrum of the ERDRP-0519 chemotype challenging, the insight gained in this study generates high confidence that function-equivalent druggable sites exist in all paramyxovirus, and potentially in all mononegavirus, polymerases. ERDRP-0519 is a clinical anti-MeV candidate that meets fundamental requirements of a valid morbillivirus therapeutic [10,58]. In addition to its direct anti-MeV impact, the compound emerged as a class-defining prototype inhibitor that has illuminated an attractive druggable site in mononegavirus L proteins, revealed a unique inhibitory mechanism of RdRP activity, and has established a foundation for future indication spectrum expansion through scaffold engineering or targeted drug discovery.

## Materials and methods

### Cells and viruses

African green monkey kidney epithelial cells (CCL-81; ATCC) stably expressing human signaling lymphocytic activation molecule (Vero-hSLAM) were maintained at 37˚C and 5% $CO_2$ in Dulbecco's modified Eagle's medium (DMEM) supplemented with 7.5% fetal bovine serum. Insect cells (SF9) were propagated in suspension using Sf-900 II SFM media (Thermo Scientific) at 28˚C. All cell lines used in this study are routinely checked for mycoplasma and microbial contamination. All transfections were performed using GeneJuice transfection reagent (Invitrogen), unless otherwise stated.

### Molecular biology

Plasmids encoding an MeV minireplicon luciferase reporter and non-codon optimized MeV N, P, and L genes under T7 promoter control were previously described [59,60]. All candidate resistance mutations were rebuilt in MeV L by site-directed mutagenesis of the expression plasmid using the QuikChange protocol (Stratagene), followed by validation of the modified codons through Sanger sequencing. For MeV minigenome experiments minigenome studies, resistance mutations were cloned into MeV-Edm L plasmids under T7 control using PCR mutagenesis. All plasmids were sequenced to confirm resistance mutations and sequence

integrity. For generation of the baculovirus protein expression system used for RdRP and BLI assays, MeV (strain IC-B) L and P genes were codon optimized and cloned into the Fastbac dual vector (ThermoFisher Scientific) as previously described [36], expressing L and P under polyhedron and P10 promotor control, respectively. A C-terminal His-tag was added to the P gene. To express and purify the MeV L fragment $L_{1708}$, the full-length L gene was replaced in the Fastback dual vector for an L fragment encoding for amino acids 1–1708 and harboring C-terminal FLAG and His tags.

## Chemical synthesis

All materials were obtained from commercial suppliers and used without purification, unless otherwise noted. Dry organic solvents, packaged under nitrogen in septum sealed bottles, were purchased from EMD Millipore and Sigma-Aldrich Co. Reactions were monitored using EMD silica gel 60 $F_{254}$ TLC plates or using an Agilent 1200 series LCMS system with a diode array detector and an Agilent 6120 quadrupole MS detector. Compound purification was accomplished by liquid chromatography on a Teledyne Isco CombiFlash RF+ flash chromatography system. NMR spectra were recorded on an Agilent NMR spectrometer (400 MHz) at room temperature. Chemical shifts are reported in ppm relative to residual $CDCl_3$-$d_6$ signal. The residual shifts were taken as internal references and reported in parts per million (ppm). To synthesize ERDRP-0519$_{az}$, precursor compound **1** (0.46 gm, 1.0 mmol) (S15 Fig) [11] was dissolved in anhydrous DMF (10 ml) in a 50 ml RBF and cooled to 0˚C. Under inert atmosphere, 60% NaH in mineral oil (0.06 gm) was added and stirred for 10 minutes. 1-azido-4-(bromomethyl)benzene (0.318 gm, 1.5 mmol) was added and continued stirring at room temperature for 5 hours. After completion, the reaction mixture was quenched with methanol and the solvent was removed under reduced pressure. The crude product was dissolved in dichloromethane (50 ml) and extracted with water (50 ml) and brine (50 ml). The organic layer was dried over anhydrous $Na_2SO_4$, filtered and concentrated under reduced pressure. The crude product was purified by flash column chromatography using ethyl acetate and hexane as eluent. Pure ERDRP-0519$_{az}$ (S15 Fig) was obtained as colorless solid, yield 67% (0.4 gm).

[1]H NMR (CDCl$_3$, 400 MHz): δ 7.78 (d, $J$ = 8 Hz, 2H), 7.19 (d, $J$ = 8 Hz, 2H), 7.06 (d, $J$ = 8 Hz, 2H), 6.94 (d, $J$ = 8 Hz, 2H), 5.79 (s, 1H), 5.04 (s, 2H), 4.21–4.16 (m, 4H), 3.84–3.64 (m, 3H), 3.03–2.96 (m, 1H), 2.76–2.73 (m, 1H), 1.99–1.91 (m, 1H),1.57–1.36 (m, 5H), 1.15–1.07 (m, 1H), 0.97–0.87 (m, 1H). [19]F NMR (376 MHz, CDCl$_3$) δ -62.31; [13]C NMR (100 MHz, CDCl$_3$) δ 159.61, 145.18, 141.56, 140.14 (q, $J$ = 38 Hz), 140.05, 136.18, 132.29, 130.28, 130.14, 128.41, 128.28, 120.45 (q, $J$ = 268 Hz), 119.45, 119.31, 107.16 (d, $J$ = 17 Hz), 58.30, 53.17, 49.54, 49.41, 40.97, 40.15, 40.02, 32.31, 27.88, 24.16, 18.31. MS (ES-API) [M+Na]$^+$: 614.0

## Minigenome dose-response assays

MeV firefly luciferase based minigenomes were performed as described [36]. BSR-T7/5 cells ($1.1 \times 10^4$ per well in a 96-well plate format) were transfected with N-Edm, P-Edm, L-Edm or L-Edm resistant mutant variants, and the MeV luciferase replicon reporter, followed by incubation with three-fold serial dilutions of compound. Luciferase activities were measured using a BioTek Synergy H1 multimode microplate reader approximately 40 hours after transfection. Raw data were normalized according to norm. value [%] = (RLU$_{sample}$—RLU$_{min}$) / (RLU$_{max}$—RLU$_{min}$)*100, with RLU$_{max}$ representing vehicle (DMSO) volume equivalent-treated transfected wells and RLU$_{min}$ representing transfected wells lacking the L polymerase-encoding plasmid. Four-parameter variable slope regression modeling was used to determine inhibitory concentrations (EC$_{50}$).

## Antiviral dose-response assays

Compound was added in three-fold dilutions to 96-well plates seeded with Vero-hSLAM cells ($1.1 \times 10^4$ per well), followed by infection (MOI = 0.2 $TCID_{50}$ units per cell) with a recombinant MeV-NanoPEST reporter virus harboring Nano luciferase [36]. After 30-hour incubation, Nano luciferase signals were quantified using a BioTek Synergy H1 multimode plate reader. Raw data were normalized according to norm. value [%] = ($RLU_{sample}$—$RLU_{min}$) / ($RLU_{max}$—$RLU_{min}$)*100, with $RLU_{max}$ representing vehicle (DMSO) volume equivalent-treated infected wells and $RLU_{min}$ representing infected wells exposed to 1 mg/ml cycloheximide. Four-parameter variable slope regression modeling was used to determine $EC_{50}$ and $EC_{90}$ concentrations.

## Cytotoxicity testing

To determine cytotoxic concentrations Vero-hSLAM cells were plated in 96 well plates ($1.1 \times 10^4$ per well) and incubated with three-fold serial dilutions of compound. After incubation of uninfected cells for 72 hours, PrestoBlue substrate (Invitrogen) was added to quantify cell metabolic activity as described [61,62]. Signal was measured using a BioTek H1 synergy multimode microplate reader. Where applicable, cytotoxic concentrations ($CC_{50}$) were calculated based on four-parameter variable slope regression modeling.

## Analysis of synergy between ERDRP-0519 and GHP-88309

To test synergy in combination therapy, Vero-hSLAM cells in 96-well plate format ($1.1 \times 10^4$ cells per well) were incubated with media (DMEM with 7.5% FBS) containing various concentrations of ERDRP-0519 and GHP-88309 and subsequently infected with MeV expressing a nano-luciferase reporter (MOI = 0.2 $TCID_{50}$ units per cell). Nano-luciferase activity was measured at 36 h post-infection as previously described [36]. For toxicity synergy, Vero-hSLAM cells in 96-well format ($1.1 \times 10^4$ cells per well) were incubated with media (DMEM with 7.5% FBS) containing various concentrations of ERDRP-0519 and GHP-88309 for 72 hours at 37°C. PrestoBlue substrate (Invitrogen) was then added to measure cell metabolic activity as previously described [35]. Raw values were normalized using the following formula: norm. value [%] = ($Signal_{Sample}$ − $Signal_{Min}$)/($Signal_{Max}$ − $Signal_{Min}$) × 100, where $Signal_{Min}$ equals the average of four positive control wells (receiving 100 μg/ml cycloheximide) and $Signal_{Max}$ is the negative control well (DMEM with 7.5% FBS and 0.01% DMSO). In order to perform synergy analysis, normalized data were used to generate synergy matrices with values ranging from 0 to 100, from which dose-responses and synergy scores were calculated with the Combenefit software package [63]. All graphic visualizations of synergy maps were generated in Combenefit.

## qRT-PCR analyses of viral and cellular transcripts

For MeV transcripts, Vero-hSLAM cells were infected with recMeV-Anc (MOI = 3 $TCID_{50}$ units/cell) by spin-inoculation (2000 x g, 30 min, 4°C) and incubated in the presence of ERDRP-0519 or vehicle at 37°C for 4 hours. Total RNA was extracted four hours after infection using Trizol and subjected to reverse transcription using either oligo-dT primers. Subsequent qPCR used primer pairs specific for MeV P mRNA, MeV H mRNA, MeV L mRNA, or human GAPDH mRNA, respectively. qPCR was performed using an Applied Biosystems 7500 Real-Time PCR System with a StepOnePlus Real-Time PCR System. Samples were normalized for GAPDH.

## MeV polymerase complex expression and purification

MeV P and L proteins were expressed in the Fastbac dual expression system in SF9 cells as previously described [36]. Approximately 76 hours after infection, cells were lysed in buffer containing 50 mM $NaH_2PO_4$, 150 mM NaCl, 20 mM imidazole, pH 7.5, 0.5% NP-40 buffer. MeV $L_{1708}$-P and MeV L-P complexes were purified by Ni-NTA affinity chromatography Protein complexes were eluted using buffer containing 50 mM $NaH_2PO_4$, pH 7.5, 150 mM NaCl, 0.5% NP-40, and 250 mM imidazole, and buffer were subsequently exchange to 150 mM NaCl, 20 mM Tris-HCl, pH 7.4, 1 mM DTT and 10% glycerol by dialysis or with Zeba desalting spin columns. Purification of RSV L was carried out as previously described [35,40]. Briefly, codon-optimized RSV strain A2 L and P proteins were co-expressed in insect cells (SF9) using the Bac-to-Bac Baculovirus Expression System. At 72 hours post-infection, cells were pelleted and gently lysed in 50 mM NaH2PO4 [pH 8.0], 150 mM NaCl, 20 mM imidazole, 0.5% NP-40 with Pierce Protease Inhibitor and Pierce universal nuclease. A C-terminal hexahistidine tag added to the P protein allowed co-purification of P-L complexes through immobilized metal affinity chromatography (IMAC). After 2 hours incubation of the lysates with HisPur Ni-NTA Resin (ThermoFisher Scientific), the resin was washed 5 times with 10 bed volumes of wash buffer (lysis buffer with imidazole increased to 60 mM), then bound proteins eluted with lysis buffer containing imidazole at 250 mM. Eluates were dialyzed with Slide-A-Lyzer Dialysis Cassettes 10K MWCO (ThermoFisher Scientific) in 20 mM Tris-HCl [pH 7.4], 150 mM NaCl, 10% glycerol, 1 mM dithiothreitol and stored at -80˚C.

## P-L sample purity and composition

The stained protein lane of the gel was excised in its entirety, reduced with TCEP, alkylated with iodoacetamide, and digested with trypsin. Resulting digests were analyzed using a 90-min LC gradient on the Thermo Q Exactive HF mass spectrometer. Protein quantification was performed using shared (Razor)+unique peptides. Razor peptides were assigned to the protein group with the most other peptides. Proteins identified by a single razor+unique peptide were considered low confident identifications and removed from the dataset.

## Biolayer interferometry

MeV L, RSV L and MeV $L_{1708}$ constructs were expressed and purified as previously described [36]. EZ-Link NHS-Biotin (ThermoFisher) was added at a 1:1 molar ratio for statistical mono-biotinylation of L preparations. Biotinylation was allowed to proceed for 15 minutes at room temperature and 2 hours on ice. Subsequently, buffer was exchanged to Octet Kinetics buffer (Fortebio) using a Zeba desalting spin column. Biotinylated MeV $L_{1708}$ preparations were then bound to Super Streptavidin (SSA) high-binding biosensors (Fortebio) and incubated with increasing concentrations of ERDRP-0519 (0.1 μM to 200 μM) with alternating incubations in Kinetics buffer to generate small-molecule binding and dissociation curves. SSA sensors loaded with mouse anti-FLAG IgG were used as reference sensors and additional biocytin blocked sensors were used as double parallel controls for nonspecific binding. Local fitting of binding kinetics was used to determine $K_D$ values using the Octet Red software package by Pall ForteBio. To visualize whether saturation of binding was reached, concentration-dependent steady-state sensor response signals were plotted.

### *In vitro* MeV RdRP assay

For *de novo* polymerase assays, purified P-L hetero-oligomers containing approximately 20 ng of L protein were diluted in 50 μl reaction buffer containing 1 mM DTT, 10% glycerol, 50 mM

Tris/Cl (pH 7.4) and mixed with 3 mM $MnCl_2$, 1 mM of GTP, UTP and CTP, 50 μM ATP, 10 μCi of α-$^{32}$P labelled ATP (Perkin Elmer), 1 μM of a 16-nt RNA template and 5% DMSO containing ERDRP-0519 at the specified concentration. The 16-nt RNA template sequence 3'-UGGUCUUUUUUGUUUC 5' (Dharmacon), was derived from the MeV Trailer complement promoter sequence with an additional UUUUU insertion that has been reported to promote polyadenylation for paramyxoviral polymerases [36,51]. Reactions were incubated at 30˚C for 5 hours, RNAs ethanol-precipitated overnight at -20˚C, resuspended in 50% deionized formamide with 10 mM EDTA, heat-denatured at 95˚C for 5 minutes, and fractionated through 20% polyacrylamide gels containing 7 M urea in Tris-borate-EDTA buffer. Polymerization products were visualized through autoradiography with either X-ray films (CL-XPosure, ThermoFisher Scientific) or exposed to BAS Storage Phosphor Screens MS 2040 and scanned on a Typhoon FLA7000 imager (GE Healthcare) for densitometric quantifications. *De novo* polymerase assays were performed on a 25-nt RNA template derived from the RSV trailer complement promoter 3´-UGCUCUUUUUUUCACAGUUUUUGAU 5' [40] (Dharmacon), using the same transcription buffer as above with the following adjustments: 4 mM $MnCl_2$, 2 μM 25-mer RNA template, 1 mM ATP, 50 μM GTP and 10 μCi of α-32P labelled GTP (Perkin Elmer).

### *In vitro* primer extension MeV RdRP assay

For primer extension assays, purified P-L hetero-oligomers containing approximately 100 ng of L protein were diluted in 5 μl reaction buffer containing 1 mM DTT, 5% glycerol, 10 mM Tris/Cl (pH 7.4) and mixed with 3 mM $MnCl_2$, 10 μM of either GTP and ATP or GTP only as specified in figure captures, 2.5 μCi of α-32P labelled GTP (Perkin Elmer), 4 μM of 16-nt RNA template, 100 μM of 5'-phosphorylated 4-nt primer ACCA (Perkin Elmer) and 5% DMSO containing ERDRP-0519 at the specified concentration. Reactions were incubated at 30˚C for 5 hours, stopped with one volume of deionized formamide with 20 mM EDTA, and analyzed through electrophoresis and autoradiography as described above.

### Photoaffinity labeling-based target mapping

Purified MeV L$_{1708}$ was incubated with 40 μM ERDRP-0519$_{az}$ for 15 minutes prior to activating the crosslinker. The MeV L1708 -ERDRP-0519$_{az}$ mixture was incubated on ice and exposed to UV light (365 nm) for 30 min. The sample was then exposed to additional UV light (254 nm) for 15 minutes. Protein was then collected using FLAG resin. Bound resin was then incubated with Laemmli buffer at 56˚C for 15 minutes. SDS-PAGE electrophoreses was then performed using Laemmli buffer on 4–15% acrylamide gels. Bands of interest were excised and analyzed by mass spectrometry. ERDRP-0519$_{az}$ crosslinked peptides were identified by the Proteomics & Metabolomics Facility at the Wistar Institute as previously described [36]. In order to identify ERDRP-0519$_{az}$ crosslinked peptides, mass addition of 563.1814 (MW of ERDRP-0519$_{az}$) was considered for all amino acid residues.

### 3D-QSAR model building

The MOE software package (MOE 2018.1001 [64]) was used to perform all conformation searches, energy minimization, and model building. 3D-QSAR modeling was performed using the AutoGPA module embedded in MOE [43], as previously reported [35]. For the creation of predictive 3D-QSAR models, a set of 33 diverse analogs reported in [11,13,65] were chosen exhibiting various inhibitory activities ($EC_{50}$ concentrations from 0.005–65 μM). This set of compounds was divided into a training set (16 entries,15a-i, 15k, 15m-p, 15r, 16677) and a test set (17 entries, 1a, 28c, 2g, 2i, 2k, 3a, 3-d, 9a, AS-105, AS-136, MS-12, MS-26, MS-27, ERDRP-

519az, and ERDRP-0519). Inhibitory concentrations ($EC_{50}$) were converted to pIC50 ($-\log_{10}(EC_{50})$). Conformational libraries of each compound were generated using the conformational search function of the AutoGPA package as previously reported [35]. A total of 2398 and 3485 conformations were generated for the training and test sets, respectively. Training set conformations were then aligned and assigned pharmacophore features using the pharmacophore elucidation function of MOE. AutoGPA identified common features and created 287 initial pharmacophore models based on the training set. After electrostatic, steric, and space filling model building was performed, a partial-least squares analysis was performed and each model was validated, scored, and ranked based on LOO correlation ($q^2$) and goodness of fit ($R^2$). The predictive potential of the model was then tested using a conformation database of the test set. Predicted pIC50 values were then compared with actual pIC50 values to determine the correlation ($R^2$) and slope of the correlation.

## Homology modeling, *in silico* ligand docking, and pharmacophore extraction

Docking studies were performed with MOE 2018.1001, using the Amber10 force field. Homology models of MeV L based on the coordinates reported for PIV-5 L (PDB 6v85), RSV-L (PDB 6pzk), VSV L (PDB 5a22), and RABV L (PDB 6ueb) were used for docking studies. For structure preparation, the Protonate 3D routine of MOE was executed, using the generalized Born/volume integral formalism for electrostatics set to 300˚K, pH 7, and 0.1 M salt, with a cutoff of 15 Å and a dielectric constant of 2. Solvent dielectric constant was set to 80 and the 800R3 option with a cutoff of 10 Å was used for van der Waals interactions. Initial docking was performed using residues H589, S768, T776, 952–966 (crosslinked peptide 1), 1151–1167 (crosslinked peptide 2), L1170, R1233, and V1239 as the target site for docking of ERDRP-0519, AS-136 [65], and 16677[13]. Target selection was based on resistance data and crosslinking results. No Wall constraints were used. Placement was performed by the triangle matcher method using London dG scoring. The top 30 poses of each compound were further refined using the induced fit method with 500 iterations of energy minimization using a 0.01 RMS gradient. All favorable docking poses were localized proximal to residues 1155–1158. These residues were selected as the target for *in silico* docking and an additional round of binding was performed for ERDRP-0519, AS-136, and 16677. Results from this docking procedure identified a favorable shared docking pose. For covalent docking of ERDRP-0519$_{az}$, residues 1155–1158 were chosen as the reactive site and an induced-fit covalent protocol was used to dock the ERDRP-0519$_{az}$ structure into MeV L.

## Statistical analysis

Source data for all numerical assays conducted in this study are provided in S9 Data. Excel and GraphPad Prism software packages were used for data analysis. One-way or two-way ANOVA with Dunnett's, multiple comparisons post-hoc test without further adjustments were used to evaluate statistical significance when more than two groups were compared or datasets contained two independent variables, respectively. The specific statistical test applied to individual studies is specified in figure legends. When calculating antiviral potency and cytotoxicity, effective concentrations were calculated from dose-response data sets through 4-parameter variable slope regression modeling, and values are expressed with 95% confidence intervals (CIs) when available. Biological repeat refers to measurements taken from distinct samples, and results obtained for each biological repeat are shown in the figures along with the exact sample size (n). For all experiments, the statistical significance level alpha was set to <0.05, exact P values are shown in individual graphs.

## Supporting information

**S1 Fig. Multiple sequence alignment of different mononegavirus L proteins.** MeV
(NP_056924.1), HPIV3 (ARA15380.1), HPIV1 (ARB07783.1), NiV (AAY43917.1), MuV
(AWI67642.1), PIV5 (YP_138518.1), RSV (YP_009518860.1), HMPV (Q6WB93.1), VSV
(Q98776.1), and EBOV (NP_066251.1) L proteins were aligned using Clustal-Omega [66].
Positions of the RdRP (cyan), capping (green), connector (yellow), MTase (orange), and C-ter-
minal (pink) domains are shown as colored bars above the sequences. Positions of ERDRP-
0519 and GHP-88309 resistance mutations are highlighted in pink and salmon, respectively.
Locations of the proposed RdRP and PRNTase motifs are labeled highlighted in purple and
blue, respectively. The residues surrounding the binding pocket of ERDRP-0519 are boxed out
(black lines).
(PDF)

**S2 Fig.** Location of ERDRP-0519 resistance mutations (black) in comparison with escape sites
from an RSV L capping inhibitor (blue) and RSV L blocker AZ-27 (magenta). Side views of
the MeV L complex and top view of the RdRP and capping domains only are shown. The
GDNQ catalytic center is highlighted in red.
(PDF)

**S3 Fig. Cross-species conservation of morbillivirus L residues involved in ERDRP-519
drug resistance.** Consensus sequences for residues identified in previous resistance profiling
studies in MeV and CDV against ERDRP-0519. All complete L protein sequences in the NCBI
virus database [67] for MeV (332 sequences), CDV (178 sequences), and peste des petits rumi-
nants virus (PPRV; 55 sequences; S10 Data) were aligned with Clustal-Omega [66] and con-
sensus sequences were generated using WebLogo [68]. The specific residue involved in
resistance to ERDRP-0519 is boxed.
(PDF)

**S4 Fig. Synergy between ERDRP-0519 and GHP-88309. A-D)** Antiviral synergy scores
mapped to the dose response of ERDRP-0519 vs GHP-88309 measuring antiviral potency
($n = 4$) or cytotoxicity ($n = 5$). Synergy scores plotted over the matrix of different compound
concentrations tested (blue = synergy, red = antagonism) are shown in (A). The sum of syn-
ergy and antagonism observed for ERDRP-0519 and GHP-88309 against MeV was positive
(26.64; $syn_{max} = 42.1$, $ant_{max} = -0.89$), indicating synergy, but drug combinations did not
increase cytotoxicity (B). Data analysis according to the HSA model [69] for antiviral (C) and
cytotoxic (D) effects. **E-F)** Matrices showing the synergy scores for antiviral efficacy (E) or
cytotoxicity (F) over a range of different compound concentrations.
(PDF)

**S5 Fig. Experimental repeats of target binding affinity of ERDRP-0519. A-C)** Experimental
repeats of BLI of ERDRP-0519 and purified standard (WT) full-lengths MeV L (A) and MeV
$L_{1708}$ (B-C). $K_D$ values and goodness of fit are shown for each construct. **D-F)** Concentration-
dependent steady-state BLI sensor response signals from results shown in (A-C) were plotted
for full-lengths L (D) and MeV $L_{1708}$ (E-F).
(PDF)

**S6 Fig. Experimental repeats of target binding affinity of ERDRP-0519 to MeV P-$L_{1708}$
complexes harboring different resistance mutations. A-C)** Experimental repeats of BLI of
ERDRP-0519 and purified MeV $L_{1708}$ resistance mutations H589Y(A), S768P (B) and L1170F
(C). $K_D$ values and goodness of fit are shown for each construct. **D-F)** Concentration-

dependent steady-state BLI sensor response signals from results shown in (A-C).
(PDF)

**S7 Fig. Steady-state analyses of BLI binding saturation shown in Fig 2. A-F)** Concentration-dependent steady-state BLI sensor response signals were plotted for the different L populations (full-length MeV L (A), MeV $L_{1708}$ (B) RSV L (C) and MeV $L_{1708}$ carrying resistance mutations H589Y(D), S768P (E) and L1170F (F).
(PDF)

**S8 Fig. Substitutions in the RdRP catalytic site inhibit RNA synthesis in the primer extension assay.** Purified recombinant WT MeV L-P complexes or complexes harboring $L_{D773A}$, $L_{N774A}$ or $L_{D773A\ N774A}$ substitutions in the highly conserved RdRP GDN motif were incubated with a 16-nt RNA template, a 5'-phosphorylated 4-nt primer, and the specified NTPs to assess primer extension. RNA products were separated by 7M urea 20% polyacrylamide gel electrophoresis and visualized by autoradiography. Representative autoradiogram is shown (n = 3). As a control for background signal due to unincorporated isotope, a reaction with no enzyme is included.
(PDF)

**S9 Fig. ERDRP-0519$_{az}$ inhibits MeV RdRP activity in the *in vitro* RdRP assay.** Purified recombinant standard MeV L-P complexes or complexes harboring substitutions in the RdRP GDN motif $L_{D773A}$, $L_{N774A}$ or $L_{D773A\ N774A}$ as specified were incubated with a 16-nt RNA template to assess the effect of ERDRP0519az on *de novo* RNA synthesis. RNA products were separated by 7M urea 20% polyacrylamide gel electrophoresis and visualized by autoradiography. A representative autoradiogram is shown (n = 2).
(PDF)

**S10 Fig. Peptides in MeV L identified as candidate targets for ERDRP-0519 binding through photo-affinity mapping with ERDRP-0519$_{az}$.** Peptides 1 and 2 are shown in pink and purple, respectively. Crosslinking-identified peptide 3 (orange spheres) is also highlighted. Residues directly engaged by ERDRP-0519$_{az}$ are shown as dark orange spheres.
(PDF)

**S11 Fig. Synthetic evolution of the ERDRP-0519 chemotype does not alter the pharmacophore. A-F)** Top scoring *in silico* docking poses (orange sticks) and 2-D ligand interaction projections are shown for the clinical candidate ERDRP-0519 (A-B), first generation lead AS-136 (C-D), and original screening hit 16677 (E-F). Conserved hydrogen bond interactions are predicted between all scaffolds and Y1155, N1285, and H1288. Locations of peptide 2 (purple sticks), of the intrusion (blue) and priming (red) loops, and of proposed PRNTase motifs (blue sticks) are shown. **G)** Stick model overlays of the docking poses of ERDRP-0519 (orange), AS-136 (red) and 16677 (blue).
(PDF)

**S12 Fig. Conservation of morbillivirus L residues surrounding the predicted ERDRP-519 binding pocket. A)** Linear schematic of the MeV L protein displaying areas surrounding the ERDRP-0519 binding pocket. **B)** Consensus sequences based on the complete L protein sequence alignments from S3 Fig complete L protein sequences in the NCBI virus database [67] for MeV (332 sequences), CDV (178 sequences), and PPRV (55 sequences, S10 Data) were generated with WebLogo [68]. The specific residues surrounding the predicted binding pocket are boxed. **C)** Spatial organization of different regions surrounding the ERDRP-0519 binding pocket. Segments are labeled by residue range.
(PDF)

**S13 Fig.** *In silico* **docking attempts of ERDRP-0519 into PIV-5, RSV and VSV L polymerases, which are all not inhibited by the compound. A-F)** Docks of ERDRP-0519 into PIV-5 L (red sticks (A-B)), RSV L (red sticks (C-D)) and VSV L (red sticks (E-F)) did not yield poses similar to that obtained for docking into MeV L (orange sticks). 2-D schematics of the top scoring docking poses are shown for each structure. Peptide 1 (pink), peptide 2 (purple), the intrusion (blue) and priming (red) loops, ERDRP-0519 resistance mutations (black spheres), and the proposed PRNTase motifs (blue spheres) are marked.
(PDF)

**S14 Fig. Conformational rearrangements of the priming and intrusion loops into initiation conformation eliminates ERDRP-0519 binding pocket. A-I)** Overlay of the ERDRP-0519 docking pose into MeV L homology models based on PIV-5 (intrusion loop down; A, D, G), RABV (intrusion loop up; B, E, H), and RSV (intrusion loop up; C, F, I) L proteins. Close up views of the ERDRP-0519 binding pocket in the different MeV L models are shown in (D-F). Steric incompatibilities (red scaffold substructures) are noted with RABV and RSV-based models (H-I) with intrusion loop in initiation conformation.
(PDF)

**S15 Fig. Chemical synthesis strategy of ERDRP-0519$_{az}$.**
(PDF)

**S1 Table. Mass spectrometry analysis of recombinant P-L complex preparations.**
(DOCX)

**S2 Table. ERDRP-0519 analogs used for development of the 3D-QSAR model.**
(DOCX)

**S1 Data. Source and biological repeats from Fig 2A.**
(PDF)

**S2 Data. Mass-spectrometry analysis of MeV P-L preparation.**
(XLSX)

**S3 Data. Source and biological repeats from Fig 3C.**
(PDF)

**S4 Data. Source and biological repeats from Fig 3D.**
(PDF)

**S5 Data. Source and biological repeats from S8 Fig.**
(PDF)

**S6 Data. Source and biological repeats from Fig 4B and 4C.**
(PDF)

**S7 Data. Source and biological repeats from Fig 4D.**
(PDF)

**S8 Data. Source and biological repeats from S9 Fig.**
(PDF)

**S9 Data. Source data of all numerical assays performed in this study.**
(XLSX)

**S10 Data. Accession numbers of morbillivirus L sequences included in multiple sequence alignments shown in S3 and S12 Figs.**
(DOCX)

## Acknowledgments

We thank H-Y Tang and the Wistar Institute Proteomics and Metabolomics Facility for assistance with proteomics analysis and KK Conzelmann for the BSR-T7/5 stable cell line.

## Author Contributions

**Conceptualization:** Robert M. Cox, Julien Sourimant, Richard K. Plemper.

**Data curation:** Robert M. Cox, Julien Sourimant.

**Formal analysis:** Robert M. Cox, Richard K. Plemper.

**Funding acquisition:** Richard K. Plemper.

**Investigation:** Robert M. Cox, Julien Sourimant, Mugunthan Govindarajan.

**Methodology:** Robert M. Cox, Richard K. Plemper.

**Project administration:** Richard K. Plemper.

**Resources:** Richard K. Plemper.

**Supervision:** Michael G. Natchus, Richard K. Plemper.

**Validation:** Robert M. Cox, Richard K. Plemper.

**Visualization:** Robert M. Cox, Julien Sourimant, Richard K. Plemper.

**Writing – original draft:** Robert M. Cox, Julien Sourimant, Richard K. Plemper.

**Writing – review & editing:** Robert M. Cox, Julien Sourimant, Mugunthan Govindarajan, Richard K. Plemper.

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
