## [Decision Letter · Decision Letter 0]

2 Nov 2020

Dear Dr. Plemper,

Thank you very much for submitting your manuscript "Therapeutic Targeting of Measles Virus Polymerase with ERDRP-0519 Suppresses All RNA Synthesis Activity" for consideration at PLOS Pathogens. As with all papers reviewed by the journal, your manuscript was reviewed by members of the editorial board and by several independent reviewers. In light of the reviews (below this email), we would like to invite the resubmission of a significantly-revised version that takes into account the reviewers' comments.

In your revision, please pay particular attention to adressing the comments regarding scientific rigor (e.g., protein purity, missing controls etc.) and addition of methodological information (in particular on molecular modeling). Additionally, please carefully and thoroughly address all of the points raised by the reviewers. 

We cannot make any decision about publication until we have seen the revised manuscript and your response to the reviewers' comments. Your revised manuscript is also likely to be sent to reviewers for further evaluation.

Sincerely,

Christiane E. Wobus

Associate Editor

PLOS Pathogens

Mark Heise

Section Editor

PLOS Pathogens

Kasturi Haldar

Editor-in-Chief

PLOS Pathogens

orcid.org/0000-0001-5065-158X

Michael Malim

Editor-in-Chief

PLOS Pathogens

orcid.org/0000-0002-7699-2064

Reviewer's Responses to Questions

**Part I - Summary**

Reviewer #1: This article by Cox and colleague presents a molecule targeting Measles L protein and its polymerase domain. The study explore through various methods including label-free biolayer interferometry, photoaffinity cross-linking, and in vitro RdRP assays of the purified MeV L-P complex the binding characteristics, physical docking site, and the molecular mechanism of action of ERDRP-0519.

The results of the article are interesting however this article is not at the publication stage yet.

This article needs serious rewriting in its introduction. A reader that is not verse in L proteins nor Mononegavirales is unable to understand what is being attempted here. Authors have to take a great deal of care to explain the complexity of the L and its catalytic motifs A to F as well as to properly present the structural information relevant in this study. Similarly the PRNTase domain with its motifs A to E.

Some aspects in the methodology needs to be clarify, controls needs to be added and some results needs to properly presented.

Reviewer #2: The manuscript “Therapeutic Targeting of Measles Virus Polymerase with ERDRP-0519 Suppresses All RNA Synthesis Activity” by Cox et a. from Plemper’s laboratory reports the mechanism of action of a promising measles inhibitor ERDRP-0519, which was developed earlier in the same laboratory. The experimental techniques used are in close resemblance with that the Plemper’s laboratory has recently published for another inhibitor, GHP-88309. The current study identifies three peptide segments photo-crosslinked to the inhibitor of which two confined to a region between RdRp and PRNTase domains. The photo-crosslinked sites and mutation data were used to identify the inhibitor-binding pocket in the cryo-em structures of rhabdovirus L-proteins of related viruses. The resistance mutations in the proposed pocket are shown to reduce the inhibitor binding whereas, the mutations such as T776A outside the pocket have a moderate impact on the inhibitor binding. The study is relevant, however, needs to address the following points.

Reviewer #3: This manuscript describes the molecular mechanism of morbillivirus polymerase inhibition by ERDRP-0519 and its binding interactions with the protein target. The same authors previously reported ERDRP-0519 as a pan-morbillivirus inhibitor. ERDRP-0519 was shown to block morbillivirus polymerase activity using cell-based minigenome assays. The authors also previously identified unique ERDRP-0519 resistance mutations associated with strong viral fitness penalty in cell culture and in vivo. In this follow up study, the authors modeled MeV L polymerase based on the PIV-5 L coordinates, allowing 3D mapping of resistance mutations indirectly revealing the putative binding site of ERDRP-0519. They demonstrated that there is no cross resistance between ERDRP-0519 and GHP-88309, confirming the hypothesis that these two compounds bind to different sites. Additionally, the authors used a biophysical method (biolayer interferometry) to measure the detrimental effect of the resistance mutations on the binding affinity of ERDRP-0519. The authors also showed direct inhibition of viral polymerase in RdRp enzymatic assays using primer extension and de novo RNA synthesis. Resistance mutations H589Y and T776A showed reduced susceptibility to compound-mediated suppression of de novo polymerase initiation, but had no effect on the inhibition of primer extension. Finally, ERDRP-0519az was used as a photoactivable analog of ERDRP-0519 for photo-affinity labeling and proteomics-based mapping of the putative target site of ERDRP-0519.

Overall, this is an interesting study containing novel results with utility for understanding both the function of MeV polymerase and the mechanism of inhibition of ERDRP-0519. The RdRp experiments are well conducted and provide important information to confirm target identification. I am somewhat less convinced by the data from the two biophysical methods (BLI and photo-affinity peptide mapping) in parts due to the low purity of the proteins and the risk of interference from contaminants and non-specific binding. I listed important control experiments and additional data needed to consolidate the biophysical results and their associated conclusions.

**Part II – Major Issues: Key Experiments Required for Acceptance**

Reviewer #1: The Materials and Methods section is at this stage incomplete. A significant part of the article and results rely on an homology model and docking procedure of which we know almost nothing about. No proper comparison between the different L protein was shown, no alignment, no sequence identity, no structural validation of the model, no docking cage information, what residues of L were involved in the docking procedure are presented.

I am sorry to say I am unable to evaluate the validity of the presented in silico results.

As everyone knows using in silico approach always deliver. To evaluate the degree of confidence of an in silico result the reader needs the proper methodology.

MeV polymerase complex expression and purification procedure is done in one step. Affinity chromatography, How pure is your sample ? How sure are you that you do not have contaminant ? Please provide a control gel of the purification.

Page 9 line 153 it is said “A catalytically defective L mutant harboring an N774A substitution in the polymerase active site confirmed that RNA synthesis was MeV P-L specific and not due to co-purified cellular contaminants.”

However Figure 3 C presents a gel that is badly cut but for which we can clearly see an enrich signal at 16nts specifically in the control lane. As the gel is not the raw data a doubt is present. I would like to see the uncut version of the gel. Also why the same control is not done in the case of primer extension Figure 3 D? it should be done.

Reviewer #2: 1. The mode of inhibition by ERDRP-0519 is proposed to be by blocking de novo initiation of RNA synthesis, however, the proposed binding site of ERDRP-0519 appears to be away from the priming loop. Can the authors be more specific on the structural impacts of the inhibitor binding? Also, why the impact of H589Y mutation in the proposed binding pocket is minimum? Have the authors checked the impact of H598A mutation to validate the binding pocket?

2. Using single-particle cryo-em, it may be feasible to experimentally find the binding of the inhibitor to an engineered pocket in a L protein of a related virus for which the structure is already determined. If not, authors need to comment on the primary hurdle to such a study.

3. Authors need to provide a detailed bioinformatics analysis of the pocket such as sequence and structure conservation of the region in L proteins of rhabdoviruses as supplemental material.

4. Do ERDRP-0519 and GHP-88309 inhibition exhibit synergy as their proposed binding sites and resistance mutations do not overlap?

5. Have the authors checked the impacts of distant mutations at 1233 and 1239 positions on inhibitor binding?

Reviewer #3: Major Issues:

Figure 2A: What is the approximate yield and purity of the proteins? What are the multiple bands on the gel? Are these viral or non-viral proteins in Fig 2A? Please provide protein identity of the major contaminants. I am concerned about the quality of the BLI data with low purity proteins. Are there published examples of high quality BLI data obtained with impure proteins?

Figure 2B: For the WT signal, the lowest concentration used for ERDRP-0519 is 100 nM, which is close to the reported Kd value. Can the authors repeat this experiment at lower concentration? This will help to improve the quality of the graph in Fig 2F. Also please show error bars from experimental repeats in panels F-I. In figure legend 2, please specify number of biological repeats (n) used to report the data. The authors need to make sure the wild-type BLI profile is robust and the result of multiple independent experiments because it is used for comparison with the mutant profiles and calculation of resistance level.

Figure 5 and Line 348: I also find it surprising that escape from ERDRP-0519 inhibition with resistance mutations is mediated by secondary structural effects rather than due to primary resistance, which is unusual for non-nucleoside polymerase inhibitors. There is a risk that the photoaffinity data is misleading the identification of the binding site due to non-specific crosslinking. Therefore, in figure 5 the authors need to make sure that the peptide sequence they identified is not the result of non-specific binding and cross-linking with ERDRP-0519az. This can be done by several ways: Does ERDRP-0519az inhibit RdRp activity by P-L complex? Please show data. If ERDRP-0519az and ERDRP-0519 bind to the same site, then adding ERDRP-0519 should outcompete ERDRP-0519az cross-linking. Can you demonstrate that this is the case to indirectly indicate that the two molecules bind to the same site?

**Part III – Minor Issues: Editorial and Data Presentation Modifications**

Reviewer #1: Page 4 line 35 paramyxovirus family does not exist Paramyxoviridae

The Materials and Methods molecular biology section , is unclear I understand the procedure was already published but a brief recap of the procedure would help the reader.

Page 11 line 214 “… part of a variable region in the viral polymerase (supporting figure S2) Figure S2 does not show that !

Figure 1B is too small and could be place in supplementary. On the other hand you need to show the catalytic sites their position relative to the different tunnels for RNA, label the motifs and keep the same reference through the whole article.

Reviewer #2: 1. A summary table listing the putative pocket residues and the impact of the mutations on inhibitor binding will be helpful.

Reviewer #3: Minor Issues:

Line 129: Why do the authors mention “mono biotinylation”? Can they provide more information on the molar ratio of protein versus EZ-Link NHS-Biotin at the labeling step? Did they measure the stoichiometry of biotin group per protein?

Is the putative binding site conserved among clinical strains of MeV and other morbilliviruses? What is the level of natural polymorphism from available sequence databases? Can the authors report this information in the manuscript?

Figure 3: Does ERDRP-0519 compete with NTP and RNA binding? Does increase NTP and RNA concentrations change IC50 value, as predicted by docking pose of ERDRP-0519 in the template RNA channel? (line 368).

PLOS authors have the option to publish the peer review history of their article (what does this mean?). If published, this will include your full peer review and any attached files.

Reviewer #1: No

Reviewer #2: **Yes: **Kalyan Das

Reviewer #3: No
---

## [Decision Letter · Decision Letter 1]

10 Feb 2021

Dear Dr. Plemper,

We are pleased to inform you that your manuscript 'Therapeutic Targeting of Measles Virus Polymerase with ERDRP-0519 Suppresses All RNA Synthesis Activity' has been provisionally accepted for publication in PLOS Pathogens.

Best regards,

Christiane E. Wobus

Associate Editor

PLOS Pathogens

Mark Heise

Section Editor

PLOS Pathogens

Kasturi Haldar

Editor-in-Chief

PLOS Pathogens

orcid.org/0000-0001-5065-158X

Michael Malim

Editor-in-Chief

PLOS Pathogens

orcid.org/0000-0002-7699-2064

Reviewer Comments (if any, and for reference):

Reviewer's Responses to Questions

**Part I - Summary**

Reviewer #3: The authors successfully addressed my questions and concerns.

**Part II – Major Issues: Key Experiments Required for Acceptance**

Reviewer #3: No remaining major issues

**Part III – Minor Issues: Editorial and Data Presentation Modifications**

Reviewer #3: (No Response)

PLOS authors have the option to publish the peer review history of their article (what does this mean?). If published, this will include your full peer review and any attached files.

Reviewer #3: No

---

## [Editor Report · Acceptance letter]

19 Feb 2021

Dear Dr. Plemper,

We are delighted to inform you that your manuscript, "Therapeutic Targeting of Measles Virus Polymerase with ERDRP-0519 Suppresses All RNA Synthesis Activity," has been formally accepted for publication in PLOS Pathogens.

Best regards,

Kasturi Haldar

Editor-in-Chief

PLOS Pathogens

orcid.org/0000-0001-5065-158X

Michael Malim

Editor-in-Chief

PLOS Pathogens

orcid.org/0000-0002-7699-2064